# Mitochondrial hydrogen peroxide positively regulates neuropeptide secretion during diet-induced activation of the oxidative stress response

Qi Jia [1] & Derek Sieburth[2 ✉]

Mitochondria play a pivotal role in the generation of signals coupling metabolism with neurotransmitter release, but a role for mitochondrial-produced ROS in regulating neurosecretion has not been described. Here we show that endogenously produced hydrogen peroxide originating from axonal mitochondria ($mtH_2O_2$) functions as a signaling cue to selectively regulate the secretion of a FMRFamide-related neuropeptide (FLP-1) from a pair of interneurons (AIY) in *C. elegans*. We show that pharmacological or genetic manipulations that increase $mtH_2O_2$ levels lead to increased FLP-1 secretion that is dependent upon ROS dismutation, mitochondrial calcium influx, and cysteine sulfenylation of the calcium-independent PKC family member PKC-1. $mtH_2O_2$-induced FLP-1 secretion activates the oxidative stress response transcription factor SKN-1/Nrf2 in distal tissues and protects animals from ROS-mediated toxicity. $mtH_2O_2$ levels in AIY neurons, FLP-1 secretion and SKN-1 activity are rapidly and reversibly regulated by exposing animals to different bacterial food sources. These results reveal a previously unreported role for $mtH_2O_2$ in linking diet-induced changes in mitochondrial homeostasis with neuropeptide secretion.

[1] PIBBS program, Keck School of Medicine, University of Southern California, Los Angeles, CA, USA. [2] Department of Physiology and Neuroscience, Zilkha Neurogenetic Institute, Keck School of Medicine, University of Southern California, Los Angeles, CA, USA. ✉email: sieburth@usc.edu

Mitochondria are complex organelles with roles in energy metabolism, cell signaling, and calcium homeostasis. In neurons and neuroendocrine cells, mitochondria are highly enriched at neurotransmitter release sites where they play a critical role in coupling metabolic demands with neurotransmitter and hormone secretion. Mitochondria supply a majority of cellular ATP, which is critical for fueling vesicle transport, release, and recycling[1]. Mitochondria are also important for regulating synaptic strength through the production of ATP locally at synapses[2], and for regulating vesicle exocytosis through cytoplasmic calcium buffering[3], and the production of metabolites[4–6]. Mitochondria are the primary cellular source of reactive oxygen species (ROS) that arise as byproducts of mitochondrial oxidative phosphorylation. At high levels, ROS can cause detrimental effects by oxidizing cellular components such as proteins, lipids and nucleic acids, and prolonged ROS exposure is linked to neuropathological conditions including ageing and neurodegenerative diseases[7]. However, at physiological levels, ROS can function as intracellular signals to regulate normal neuronal functions including neuronal homeostasis[8], plasticity[9–12], and activity[13–17].

Hydrogen peroxide ($H_2O_2$) is an endogenous ROS produced by mitochondria through the dismutation of superoxide ($O_2{\cdot}^-$) by mitochondrial superoxide dismutase (SOD2), and also outside the mitochondria by cytoplasmic oxidases[18,19]. $H_2O_2$ is relatively stable and freely diffusible, and in neurons mitochondrial $H_2O_2$ levels are tightly regulated by synaptic activity and by cellular antioxidant systems such as the peroxiredoxin/thioredoxin system[20,21]. $H_2O_2$ signaling has been implicated in regulating synaptic plasticity[22–25] and neurotransmitter release[26–30], and altering $H_2O_2$ levels by over-expression of superoxide dismutase causes defects in hippocampal LTP and learning paradigms in mice[31–33]. Studies of insulin secretion in pancreatic beta cells have shown that glucose-stimulated insulin secretion, which is positively regulated by ATP, is inhibited by the removal of $H_2O_2$[34,35]. Despite evidence for roles for $H_2O_2$ signaling in regulating DCV secretion from neuroendocrine cells, the subcellular source(s) and mechanism of action of $H_2O_2$ in regulating transmitter secretion are poorly understood.

Neuropeptides are conserved modulators of behavior, physiology, and homeostasis. Studies in *Caenorhabditis elegans* have established important roles for neuropeptide release from neurons in activating stress responses in distal tissues[36–38]. For example, neuropeptide secretion from specific neurons regulates the mitochondrial unfolded protein response (UPR$^{mt}$) and the heat shock response in the intestine through an inter-tissue signaling mechanism[38–40]. In addition, a number of neuropeptide signaling pathways are linked with sensory regulation of fat metabolism[41,42], and longevity[43]. Neuropeptides are packaged into dense core vesicles (DCVs) in neuronal somas, and DCVs are subsequently trafficked to release sites where they undergo soluble NSF attachment protein receptor (SNARE) and calcium-dependent exocytosis. The release properties of DCVs and synaptic vesicles (SVs) differ in several important respects including differences in release sites, calcium dependence, and release kinetics[44]. Unlike SVs, the mechanisms underlying the regulation of DCV release are not well defined.

Here we identify a role for endogenously produced $H_2O_2$ in promoting the secretion of the FMRFamide-related neuropeptide, FLP-1, from a pair of interneurons (AIY), to activate the antioxidant response in distal tissues in *C. elegans*. $H_2O_2$ originating from axonal mitochondria selectively regulates FLP-1 release, and the mitochondrial peroxiredoxin-thioredoxin antioxidant system in AIY functions to negatively regulate FLP-1 release. Changes in the bacterial diet of *C. elegans* lead to rapid changes in endogenous $H_2O_2$ levels in AIY interneurons, FLP-1 secretion, and

antioxidant response activation in distal tissues. Finally, the effects of $H_2O_2$ on neuropeptide secretion rely on a putative reactive cysteine residue in the calcium-independent protein kinase C family member, PKC-1. Our results suggest that $H_2O_2$-induced neuropeptide secretion is a mechanism by which organisms can rapidly and reversibly control antioxidant activity in response to changing environmental inputs.

## Results

**Neuropeptide signaling activates the intestinal oxidative stress response.** Nrf2 and its *C. elegans* ortholog, SKN-1, are evolutionarily conserved transcription factors that are master regulators of the antioxidant response that control the expression of antioxidant enzymes in response to oxidative stress. Nrf2 and SKN-1 are activated by ROS that originate cell autonomously[45,46], and can also be activated cell-nonautonomously by ROS originating from neighboring neurons undergoing oxidative stress[47–49]. To determine the mechanisms underlying the neuronal control of SKN-1 activation, we first sought to identify the neuronal signal(s) that promotes the antioxidant response. Juglone is a naturally occurring mitochondrial toxin that generates superoxide anion radicals[50,51] and juglone treatment of *C. elegans* increases ROS levels[52], and activates the SKN-1-dependent antioxidant response[53]. Acute treatment with juglone for 4 h leads to mild toxicity in adults 24 h later, and loss of *skn-1* significantly increases the toxicity of juglone treatment (Fig. 1a and[54,55]). We found that mutants with impaired biosynthesis or secretion of the classical neurotransmitter acetylcholine (*unc-17*/VChAT), GABA (*unc-25*/GAD), or glutamate (*eat-4*/VGAT), or the biogenic amine serotonin (*tph-1*/TPH), dopamine (*cat-2*/TH), or octopamine (*tdc-1*/DDC or *tbh-1*/DBH) exhibited similar survival following juglone treatment as wild-type controls. However, mutants with impaired neuropeptide biogenesis (*egl-3*/PC2, *egl-21*/CPE, or *sbt-1*/7B2), or secretion (*pkc-1*/PKC) exhibited significantly decreased survival following juglone treatment (Fig. 1a). *egl-3* encodes prohormone convertase 2, which performs the first cleavage step in the maturation of neuropeptide precursors into bioactive peptides in DCVs[56,57]. *egl-3* mutants cultured in media containing juglone exhibited significantly decreased survival compared to wild-type controls over all time points examined (Fig. 1b). *egl-3* mutants also exhibited decreased survival following exposure to the SKN-1-activating oxidants thimerosal or sodium arsenite, but were not sensitive to toxicity cause by heat stress (Supplementary Fig. 1a–c[58–60])

Upon activation, SKN-1 translocates into the nucleus of intestinal cells where it regulates the expression of antioxidant genes, including *gst-4*/glutathione-S-transferase[61]. *egl-3* mutants were defective in juglone-induced SKN-1::GFP nuclear translocation and juglone-induced increases in intestinal expression of the P*gst-4::gfp* reporter (Fig. 1c, d). *egl-3* mutants were not defective in the juglone-induced activation of the lifespan extension transcription factor DAF-16[54] or the unfolded protein response reporter P*hsp-4::gfp* (Supplementary Fig. 1e, f[62]). Restoring *egl-3* cDNA expression selectively in a subset of neurons (under the *flp-1* promoter, see below) fully rescued the juglone survival and juglone-induced P*gst-4::gfp* expression defects of *egl-3* mutants, whereas expressing *egl-3* cDNA in the intestine failed to rescue juglone survival (Fig. 1d, e). Together these results reveal a specific role for neuropeptide signaling originating from the nervous system in activating SKN-1 in the intestine and the antioxidant response.

**The FLP-1 neuropeptide activates the intestinal oxidative stress response.** To identify the neuropeptide(s) involved in SKN-1 activation, we first conducted a pilot RNA interference (RNAi) screen (in an *eri-1; lin-15b* background to enhance RNAi

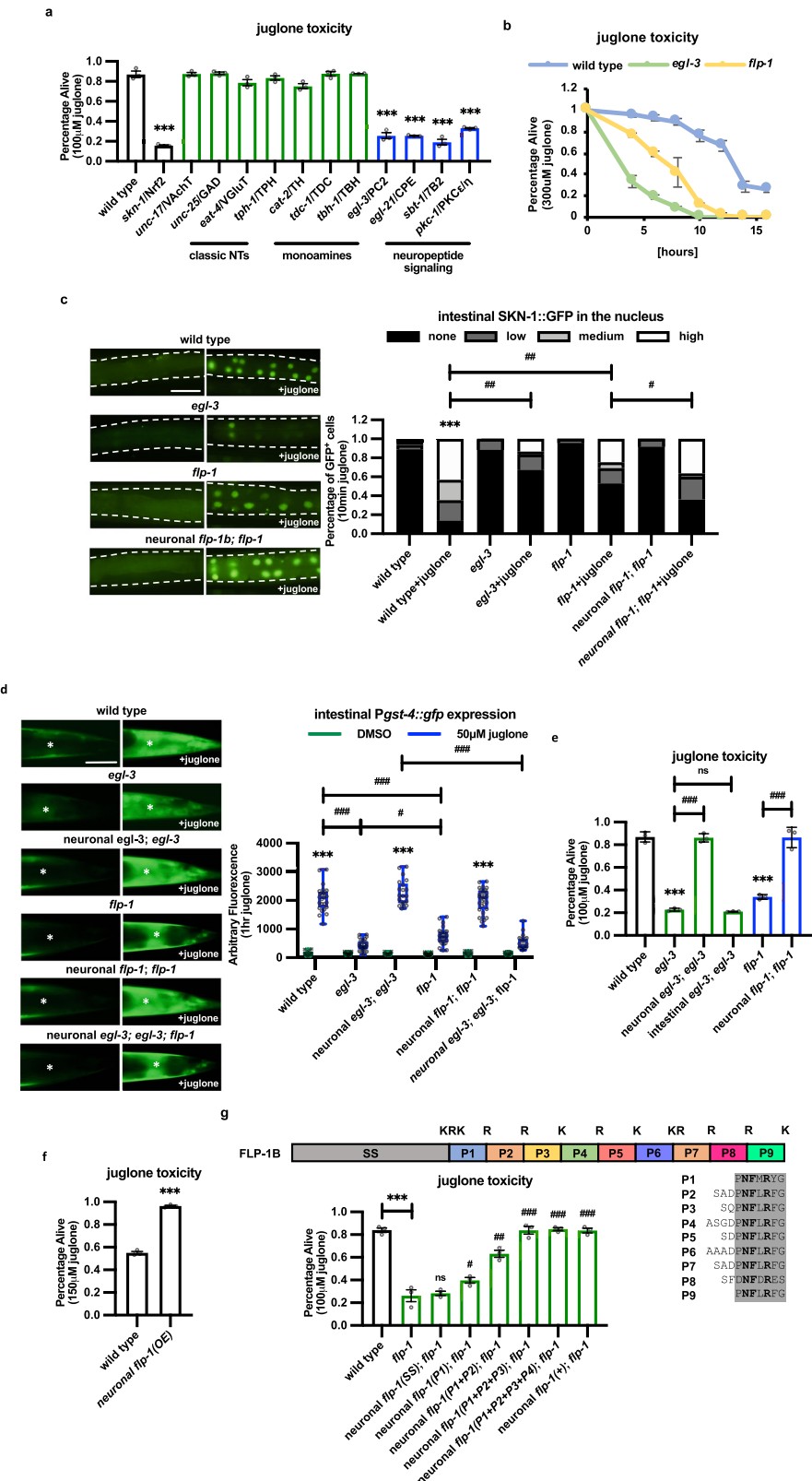

knockdown of neuronal genes[63,64]) of 88 candidates among the 111 neuropeptide-like genes encoded by *C. elegans* for altered sensitivity to toxicity by juglone (Table S1). Knockdown of six neuropeptide genes significantly increased sensitivity to juglone toxicity compared to empty vector controls (Table S1). Among these, the FMRFamide-related peptide *flp-1* emerged as a strong candidate because putative null *flp-1(ok2811)* mutants exhibited

significantly increased sensitivity to juglone-mediated toxicity compared to wild-type controls (Fig. 1b, e). *flp-1* mutants were also significantly more sensitive to toxicity caused by the oxidants thimerosal and sodium arsenite, but were not sensitive to toxicity caused by heat stress or activation of the unfolded protein response (Supplementary Fig. 1a–d). *flp-1* mutations significantly attenuated juglone-induced SKN-1::GFP nuclear translocation

**Fig. 1 Neuronal FLP-1 signaling promotes the SKN-1-mediated intestinal antioxidant response. a** Percentage of surviving animals of the indicated genotypes 16 h following 4 h treatment of young adults with juglone. VAChT vesicular acetylcholine transporter, GAD glutamate decarboxylase, VGluT vesicular glutamate transporter, TPH glutamate decarboxylase, TH tyrosine hydroxylase, TDC tyrosine decarboxylase, TBH tyramine-beta hydroxylase, PC2 prohormone convertase, SPE carboxypeptidase, 7B2 secretogranin V, PKC protein kinase C. Data are mean values ± s.e.m. n = 129, 64, 164, 108, 139, 91, 158, 165, 153, 149, 114, 63, 109 biologically independent samples over three independent experiments. ***P < 0.001 by one-way ANOVA with Dunnett's test. **b** Percentage of surviving animals of the indicated genotypes following chronic exposure of young adults to culture plates containing juglone for the indicated times. Data are mean values ± s.e.m. n = 111, 142, 121 biologically independent samples over three independent experiments. **c** Representative fluorescent images and quantification of the number of intestinal nuclei with SKN-1::GFP in adult animals following 10 min of vehicle (DMSO) or juglone treatment. Neuronal flp-1 denotes transgenes expressing flp-1 genomic DNA under control of the endogenous flp-1 promoter. Dotted lines demarcate intestinal regions. Nuclear translocation of SKN-1::GFP was measured by counting the number of fluorescent nuclei in the intestine. Fewer than 10, between 11 and 20, and above 20 fluorescent nuclei are denoted Low, Medium, and High, respectively. n = 63, 51, 52, 67, 45, 81, 36, 55 biologically independent samples. ***P < 0.001 by Student's two-tailed t-test. #P < 0.05, ##P < 0.01 by one-way ANOVA with Dunnett's test. Scale bar: 100 μm. **d** Representative images and quantification of the posterior regions of transgenic worms expressing the oxidative stress reporter Pgst-4::gfp after 1 h of juglone or vehicle (DMSO) treatment and 4 h of recovery. Asterisks mark the intestinal region used for quantitation. Pgst-4::gfp expression in body wall muscles, which appears as fluorescence on the edge of animals in some images, was not quantified. Neuronal egl-3 and neuronal flp-1 denote transgenes in which the flp-1 promoter drives expression of egl-3a cDNA and flp-1b genomic DNA, respectively. The boxes span the interquartile range, median is marked by the line and whiskers indicate the minimum and the maximum values. n = 20, 25, 30, 31, 20, 20, 31, 30, 17, 33, 19, 25 biologically independent samples. ***P < 0.001 by Student's two-tailed t-test. #P < 0.05, ###P < 0.001 by one-way ANOVA with Dunnett's test. Scale bar: 50 μm. **e** Percentage of surviving animals of the indicated genotypes 16 h following 4 h juglone treatment of young adults. Neuronal egl-3; egl-3 denotes egl-3 mutants expressing egl-3 cDNA under control of the flp-1 promoter. Intestinal egl-3; egl-3 denotes egl-3 mutants expressing egl-3 cDNA under control of the ges-1 promoter. Neuronal flp-1; flp-1 denotes flp-1 mutants expressing flp-1 genomic DNA under control of the flp-1 promoter. Data are mean values ± s.e.m. n = 89, 94, 83, 119, 129, 118 biologically independent samples over three independent experiments. ***P < 0.001 by Student's two-tailed t-test. ###P < 0.001, n.s not significant by one-way ANOVA with Dunnett's test. **f** Percentage of surviving animals of the indicated genotypes 16 h following treatment of young adults with juglone for 4 h. flp-1 (OE) denotes wild-type animals expressing flp-1b genomic DNA under control of the flp-1 promoter (Pflp-1::flp-1) transgenes as multicopy arrays. Data are mean values ± s.e.m. n = 152, 186 biologically independent samples over three independent experiments. *** P < 0.001 by Student's two-tailed t-test. **g** Top: Protein structure of FLP-1b with the predicted EGL-3 cleavage sites (K = lysine, R = arginine) marked above each mature peptide (numbered P1-P9). Below left: Percentage of surviving animals of the indicated genotypes 16 h following treatment of young adults with juglone for 4 h. Neuronal flp-1 denotes transgenes expressing the indicated flp-1 truncations under control of the flp-1 promoter. Below, right: amino acid sequence alignment of each mature FLP-1 peptide. Conserved amino acids are highlighted in gray and invariant amino acids are marked in bold. Data are mean values ± s.e.m. n = 160, 160, 159, 123, 154, 147, 90, 99 biologically independent samples over three independent experiments. ***P < 0.001 by Student's two-tailed t-test. #P < 0.05, ##P < 0.01, ###P < 0.001 by one-way ANOVA with Dunnett's test.

and Pgst-4::gfp expression, without detectably altering baseline SKN-1 activity in the absence of stress (Fig. 1c, d). The defects in survival and SKN-1 activation of flp-1 mutants were fully rescued by expression of flp-1 genomic DNA under control of its endogenous promoter fragment (Pflp-1::flp-1, Fig. 1e). In addition, the rescue of the juglone-induced Pgst-4::gfp expression defects of egl-3 mutants by egl-3 transgenes was completely blocked by flp-1 mutations (Fig. 1d). To further confirm that flp-1 is an activator of the oxidative stress response, we generated a flp-1 over-expression strain (under its own promoter) and found that these animals were significantly more resistant to juglone toxicity than non-transgenic controls (Fig. 1f). flp-1 overexpression also increased juglone-induced intestinal Pgst-4::gfp expression without altering Pgst-4::gfp expression in the absence of stress (Supplementary Fig. 1g). Together these results reveal that FLP-1 functions in the nervous system to activate the oxidative stress response by specifically regulating stress-induced SKN-1 activation in the intestine. Because the defects in juglone responsiveness of flp-1 mutants were significantly less severe than those observed in egl-3 mutants (Fig. 1b–e), additional neuropeptides processed by EGL-3 are likely to contribute to the antioxidant response.

FLP-1 is predicted to be processed into nine mature peptides that share a common C-terminal seven amino acid consensus motif (Fig. 1g[65]). Truncated flp-1 transgenes that encode just one peptide (FLP-1(P1)), two peptides (FLP-1(P1 + P2)) or, at least three peptides (FLP-1(P1 + P2 + P3), partially rescued, nearly completely rescued, or fully rescued the sensitivity to juglone-mediated toxicity of flp-1 mutants, respectively (Fig. 1g). These results indicate that the mature FLP-1 peptides may be functionally equivalent, suggesting that FLP-1 peptide levels rather than their specific sequences may determine the extent to which FLP-1 confers stress protection.

**FLP-1 secretion from the AIY interneuron activates the oxidative stress response.** flp-1 is expressed in a subset of head interneurons, and has been reported to regulate lipid homeostasis, reproduction, and behavioral plasticity in response to changes in a number of environmental cues[66–70]. We investigated in which neuron(s) flp-1 functions to activate the oxidative stress response. FLP-1 reporters are expressed in six pairs of interneurons (AVK, AVA, AVE, RIG, AIY, and AIA), and two pairs of motoneurons (RMG and M5)[71]. Several results indicate that FLP-1 secretion from the AIY interneurons (hereafter referred to as AIY) positively regulates the oxidative stress response. First, we used mito-miniSOG, which is a mitochondrially targeted light-activated singlet oxygen generator[72], to genetically ablate flp-1 expressing cells. We found that AIY ablation using four different promoters that each drive expression in AIY decreased survival following juglone treatment to a similar extent as flp-1 mutations, whereas ablation of other flp-1-expressing cells had no effect on survival (Fig. 2a). Specifically, animals expressing mito-miniSOG selectively in AIY (using the ttx-3 promoter[73], which we hereafter used to drive AIY expression), exhibited normal sensitivity to juglone toxicity in the absence of blue light, but blue light illumination of adults for 30 min, which resulted in efficient AIY ablation (Supplementary Fig. 2a), led to a significant increase in sensitivity to juglone toxicity 24 h later (Fig. 2a). Second, expression of tetanus toxin (TeTx), which disrupts SNARE-mediated exocytosis[74], in AIY conferred significant sensitivity to juglone toxicity compared to non-transgenic controls (Fig. 2b). Third, the juglone sensitivity of egl-3 mutants was rescued by expressing egl-3 selectively in AIY, and rescue was abolished by flp-1 mutations (Fig. 2c). Finally, expressing flp-1 selectively in AIY fully rescued the sensitivity to juglone toxicity of flp-1 mutants (Fig. 2d). AIY has been shown to integrate thermal and chemical cues from sensory

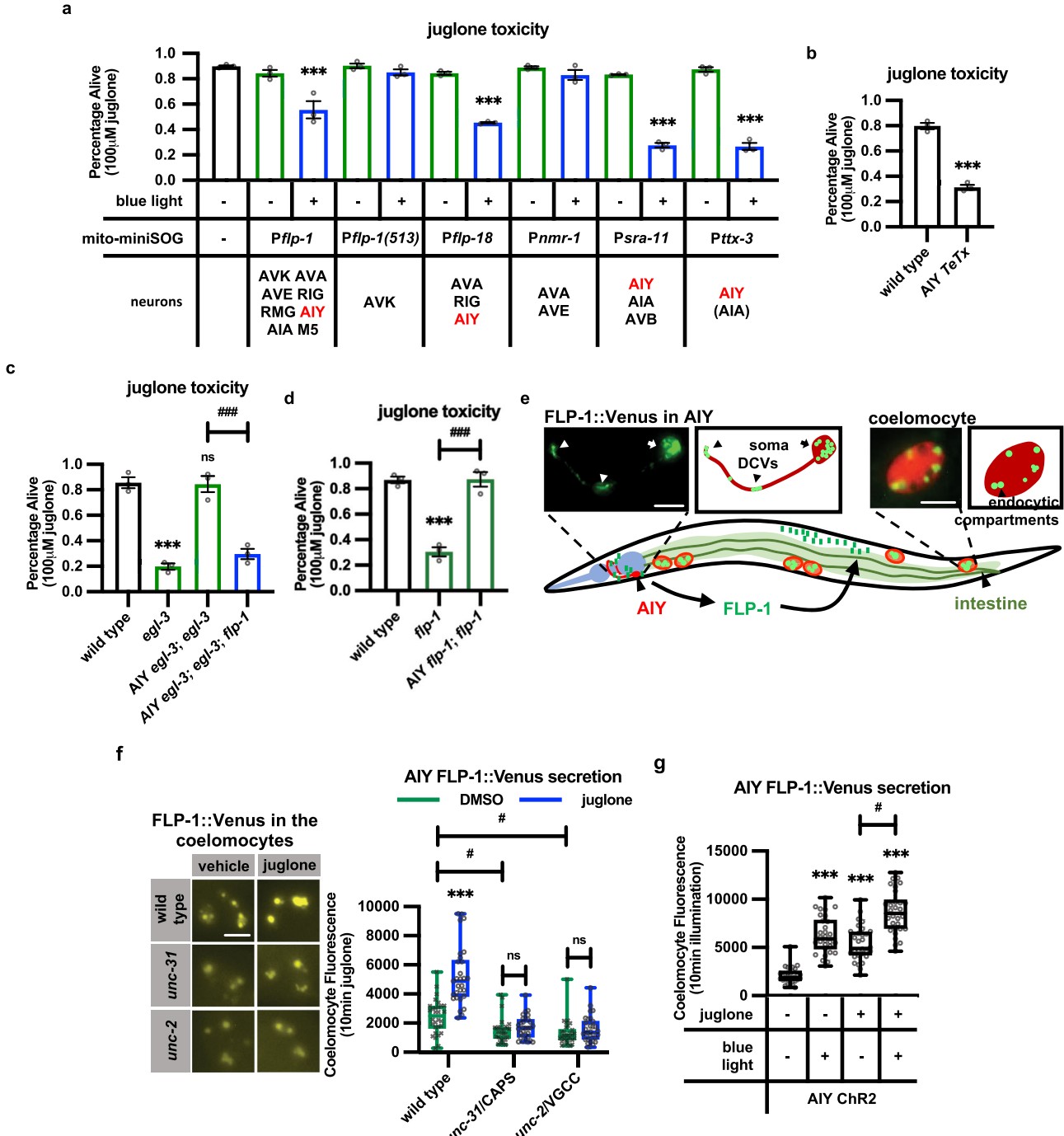

neurons to regulate behavioral and metabolic plasticity[75]. Our results reveal a previously undescribed function for AIY in regulating the oxidative stress response in peripheral tissues.

To determine whether FLP-1 is secreted from AIY, we generated transgenic animals expressing FLP-1::Venus fusion proteins in AIY. AIY is a unipolar neuron located in the head that extends a single axon anteriorly from the soma, where it forms cholinergic synapses to multiple targets in the nerve ring. FLP-1::Venus adopted a punctate pattern of fluorescence in the AIY soma and in the axon. In axons, puncta were sparsely distributed and were most prominent at the axonal bend and tip (Fig. 2e), where cholinergic presynaptic terminals are located[76,77]. FLP-1::Venus is secreted from AIY because fluorescence was also observed in coelomocytes (marked with mCherry, Fig. 2e, f), which are scavenger cells that take up material released into the

pseudocoelom into endocytic compartments by bulk endocytosis. Changes in steady-state fluorescence intensity in coelomocytes is widely used as a measure of efficacy of neuropeptide secretion[78–80]. Venus fluorescence intensity in coelomocytes of FLP-1::Venus expressing animals was significantly reduced by unc-31/CAPS or by unc-2/voltage gated calcium channel mutations, which impair calcium-dependent DCV exocytosis (Fig. 2f[80,81]). On the other hand, AIY-specific activation of the light-gated cation channel, channelrhodopsin 2 (ChR2, which increases transmitter release[82]) significantly increased FLP-1::Venus coelomocyte intensity (Fig. 2g). FLP-1::Venus coelomocyte fluorescence was significantly reduced by disruption of electron transport chain function specifically in AIY (gas-1 or mev-1 mutations[83,84]) or glycolysis (gpd-3 mutations[85]) (Supplementary Fig. 2b, c), in agreement with reports that ATP generated by

**Fig. 2 FLP-1 released from AIY interneurons promotes the antioxidant response. a** Juglone-induced toxicity of wild-type animals expressing mito-miniSOG under the control of the indicated promoters, with and without 50 mW/cm$^2$ blue light illumination for 30 min. The neurons in which each promoter drives mito-miniSOG expression are indicated below. The *ttx-3* promoter is reported to be expressed weakly in AIA neurons[190], indicated by parenthesis. Data are mean values ± s.e.m. $n = 102, 96, 98, 70, 146, 67, 77, 88, 84, 70, 85, 78, 134$ biologically independent samples over three independent experiments. ***$P < 0.001$ by Student's two-tailed *t*-test. **b** Percentage of surviving animals expressing tetanus toxin under the *ttx-3* promoter (AIY *TeTx*) 16 h following 4 h juglone treatment of young adults. Data are mean values ± s.e.m. $n = 203, 303$ biologically independent samples over three independent experiments. ***$P < 0.001$ by Student's two-tailed *t*-test. **c** Transgenic expression of *egl-3* cDNA in AIY interneurons rescued juglone-induced toxicity in *flp-1* dependent manner. The *ttx-3* promoter was used to drive the expression of *egl-3* cDNA in AIY neurons (AIY *egl-3*). Data are mean values ± s.e.m. $n = 156, 190, 162, 156$ biologically independent samples over three independent experiments. ***$P < 0.001$, n.s not significant by Student's two-tailed *t*-test. ###$P < 0.001$ by one-way ANOVA with Dunnett's test. **d** Expression of *flp-1* genomic DNA using the *ttx-3* promoter (AIY::*flp-1*) rescued juglone-induced toxicity of *flp-1* mutants. Data are mean values ± s.e.m. $n = 189, 174, 148$ biologically independent samples over three independent experiments. ***$P < 0.001$ by Student's two-tailed *t*-test. ###$P < 0.001$ by one-way ANOVA with Dunnett's test. **e** Schematic showing the positions of AIY, intestine, and coelomocytes. Transgenic animals used to measure FLP-1 secretion co-express FLP-1::Venus in AIY and mCherry in coelomocytes. An image of AIY shows the distribution of FLP-1::Venus fusion proteins in puncta in an AIY axon and soma. Scale bar: 10 μm. FLP-1::Venus secreted from AIY accumulates in the pseudocoelom and is taken up by coelomocytes. An image of the posterior-most coelomocyte that has taken up Venus into endocytic compartments is shown. DCV dense core vesicle. Scale bar: 5 μm. **f** Representative images and quantification of average coelomocyte fluorescence of the indicated mutants expressing FLP-1::Venus fusion proteins in AIY following vehicle (DMSO) or juglone treatment for 10 min. CAPS calcium-dependent activator protein for secretion, VGCC voltage-gated calcium channel. The boxes span the interquartile range, median is marked by the line and whiskers indicate the minimum and the maximum values. $n = 30$ biologically independent samples. ***$P < 0.001$, n.s not significant by Student's two-tailed *t*-test. #$P < 0.05$ by one-way ANOVA with Dunnett's test. Scale bar: 5 μm. **g** Quantification of average coelomocyte fluorescence of animals co-expressing FLP-1::Venus fusion proteins and channelrhodopsin (ChR2) in AIY with and without 1 min blue light exposure or 10 min juglone treatment. The boxes span the interquartile range, median is marked by the line and whiskers indicate the minimum and the maximum values. $n = 30$ biologically independent samples. ***$P < 0.001$ by Student's two-tailed *t*-test. #$P < 0.05$ by one-way ANOVA with Dunnett's test.

oxidative phosphorylation and glycolysis are critical for DCV secretion[86,87]. Together these results show that FLP-1::Venus is secreted from AIY via calcium- and ATP-dependent exocytosis of DCVs.

**FLP-1 antioxidant signaling is mediated by the NPR-4 GPCR in the intestine.** Neuropeptides exert their biological effects by binding to G-protein coupled receptors (GPCRs) on target cells. We reasoned that we could identify the FLP-1 receptor by screening for GPCRs that cause sensitivity to juglone toxicity when knocked down. *frpr-7* and *npr-6* are neuropeptide GPCRs reported to function downstream of FLP-1 to regulate movement[68]. We found that null *frpr-7* or *npr-6* mutants exhibited normal sensitivity to juglone toxicity (Table S2), suggesting that FLP-1-mediated activation of the oxidative stress response occurs through a different GPCR. We conducted an RNAi screen of an additional 21 GPCRs that are reported to be expressed in the intestine for altered juglone sensitivity (Table S2), and we subsequently confirmed that null mutations in *npr-4*/GPCR, significantly decreased survival following juglone-treatment compared to wild-type controls (Fig. 3a). *npr-4* was previously implicated in regulating foraging behavior, food preference, and fat homeostasis[42,69,88,89], and *npr-4* mediates the effects of *flp-1* in fat regulation[69]. The increased sensitivity to juglone-induced toxicity of *npr-4* mutants was similar to that of *flp-1* mutants, and *flp-1; npr-4* double mutants exhibited similar sensitivity to juglone as *flp-1* or *npr-4* single mutants (Fig. 3a and Table S2). The resistance to juglone-induced toxicity by *flp-1* overexpression in AIY was abolished by *npr-4* mutations (Fig. 3b). These results suggest that NPR-4 functions downstream of FLP-1 in a common genetic pathway to promote survival.

*npr-4* is reported to be expressed in the nervous system, coelomocytes, and the intestine[89]. Transgenic expression of *npr-4* cDNA in the intestine (using the *ges-1* promoter) fully restored protection against juglone-induced toxicity, whereas expression of *npr-4* cDNA in either the coelomocytes (using the *ofm-1* promoter) or nervous system (using the *rab-3* promoter) did not rescue (Fig. 3c). *npr-4* mutants had defects in juglone-induced P*gst-4*::*gfp* intestinal expression that were similar to those of *flp-1* mutants, and were not enhanced by *flp-1* mutations. The *gst-4*

expression defects of *npr-4* mutants were fully rescued by intestinal *npr-4* cDNA expression (Fig. 3d). Finally, *npr-4* mutations also blocked the enhanced juglone-induced P*gst-4*::*gfp* expression resulting from AIY *flp-1* overexpression (Fig. 3b). Together these data support a model whereby neuronal FLP-1 secretion regulates SKN-1 activation and the oxidative stress response through the activation of NPR-4 in the intestine.

**FLP-1 secretion is positively regulated by mtH$_2$O$_2$.** We hypothesized that if FLP-1 functions to activate the antioxidant response, FLP-1 secretion from AIY might be regulated by mtROS themselves (Fig. 4a). To monitor ROS levels in AIY, we used the H$_2$O$_2$ sensor HyPer, which is a dual ratio-metric sensor that increases fluorescence in response to levels of H$_2$O$_2$ as low as 0.1 μM[90,91]. HyPer targeted to mitochondria in AIY (P*ttx-3*:: mito-HyPer) adopted a punctate pattern of fluorescence in axons that colocalized with the mitochondrial marker TOMM-20:: mCherry (Supplemenraty Fig. 3a). Juglone treatment for 10 min increased mito-HyPer punctal fluorescence intensity 2.5-fold, without altering TOMM-20::mCherry intensity (Fig. 4B and Supplementary Fig. 3b). *sod-2* encodes the neuronal mitochondrial superoxide dismutase2 (SOD2), which converts superoxide anions into H$_2$O$_2$[92]. *sod-2* mutants exhibited similar mito-HyPer punctal fluorescence intensity as wild-type controls. However, juglone treatment failed to increase mito-HyPer fluorescence intensity in *sod-2* mutants (Fig. 4b). These results confirm that mito-HyPer is a specific sensor for mtH$_2$O$_2$, and they indicate that juglone treatment increases mitochondrial H$_2$O$_2$ levels in AIY axons through the dismutation of superoxide by SOD-2.

We found that juglone treatment caused a dose-dependent increase in coelomocyte fluorescence in FLP-1::Venus-expressing animals that was maximal at 300 μM (Supplementary Fig. 3c), the same concentration used for toxicity assays. A time course revealed that juglone treatment for as little as 2 min caused a significant increase in coelomocyte fluorescence in FLP-1::Venus-expressing animals and 10 min treatment resulted in a maximal response (Supplementary Fig. 3d). Similarly, a 10 min treatment with sodium arsenite significantly increased FLP-1::Venus secretion compared to untreated controls (Supplementary Fig. 3e). Juglone treatment did not detectably alter the distribution FLP-1::

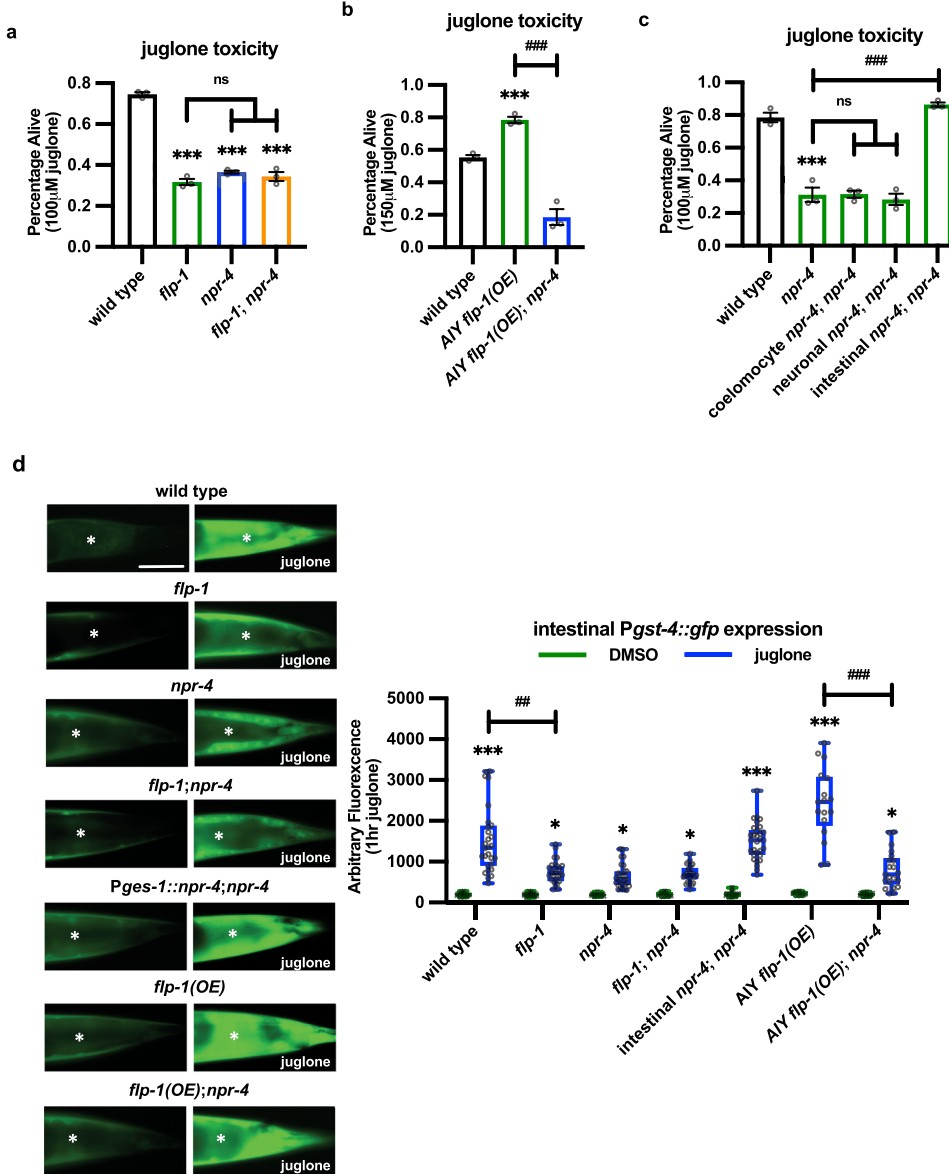

**Fig. 3 FLP-1 activates the antioxidant response through NPR-4/GPCR in the intestine. a** Percentage of surviving animals of the indicated genotypes 16 h following 4 h juglone treatment of young adults. Data are mean values ± s.e.m. n = 323, 253, 284, 301 biologically independent samples over three independent experiments. ***P < 0.001 by Student's two-tailed t-test. n.s not significant by one-way ANOVA with Dunnett's test. **b** Percentage of surviving animals of the indicated genotypes 16 h following treatment of young adults with juglone for 4 h. AIY flp-1(OE) denotes wild-type animals expressing flp-1 genomic DNA in AIY under the ttx-3 promoter (Pttx-3::flp-1). Data are mean values ± s.e.m. n = 399, 184, 211 biologically independent samples over three independent experiments. ***P < 0.001 by Student's two-tailed t-test. ###P < 0.001 by one-way ANOVA with Dunnett's test. **c** NPR-4 signaling in the intestine is required for promoting protection against juglone-induced toxicity. Transgenic expression of npr-4a cDNA in coelomocytes was driven by the ofm-1 promoter, transgenic expression of npr-4a cDNA in neurons was driven by the rab-3 promoter, transgenic expression of npr-4a cDNA in the intestine was driven by the ges-1 promoter. Data are mean values ± s.e.m. n = 144, 215, 178, 188, 212 biologically independent samples over three independent experiments. ***P < 0.001 by Student's two-tailed t-test. ###P < 0.001, n.s not significant by one-way ANOVA with Dunnett's test. **d** Representative images and quantification of the posterior regions of transgenic worms expressing Pgst-4::gfp after 1 h of vehicle (DMSO) or juglone treatment and 4 h recovery. Asterisks mark the intestinal region used for quantitative analysis. Pgst-4::gfp expression in body wall muscles, which appears as fluorescence on the edge of animals in some images, was not quantified. Intestinal rescue denotes npr-4a cDNA expressed under the ges-1 promoter (Pges-1::npr-4a). flp-1(OE) denotes animals expressing flp-1 genomic DNA in AIY under the ttx-3 promoter (Pttx-3::flp-1). The boxes span the interquartile range, median is marked by the line and whiskers indicate the minimum and the maximum values. n = 20, 28, 25, 32, 19, 29, 17, 32, 22, 31, 15, 17, 19, 21 biologically independent samples. Scale bar: 50 μm. *P < 0.05, ***P < 0.001 by Student's two-tailed t-test. ##P < 0.01, ###P < 0.001 by one-way ANOVA with Dunnett's test.

Venus fluorescence in AIY axons or somas (Supplementary Fig. 3f). sod-2 mutants, which exhibited slightly reduced FLP-1::Venus secretion, exhibited no increase in FLP-1::Venus secretion following juglone treatment (Fig. 4c). Transgenes expressing wild-type sod-2 cDNA specifically in AIY fully restored juglone-induced FLP-1 secretion to sod-2 mutants, whereas transgenes expressing sod-2 variants lacking their mitochondrial localization signal (MLS), which disrupted the mitochondrial localization of SOD-2::GFP fusion proteins (Supplementary Fig. 3g), failed to restore juglone responsiveness to sod-2 mutants (Fig. 4c). unc-31/CAPS mutations, which blocked the juglone-induced increase in FLP-1::Venus secretion (Fig. 2g), did not block juglone-induced

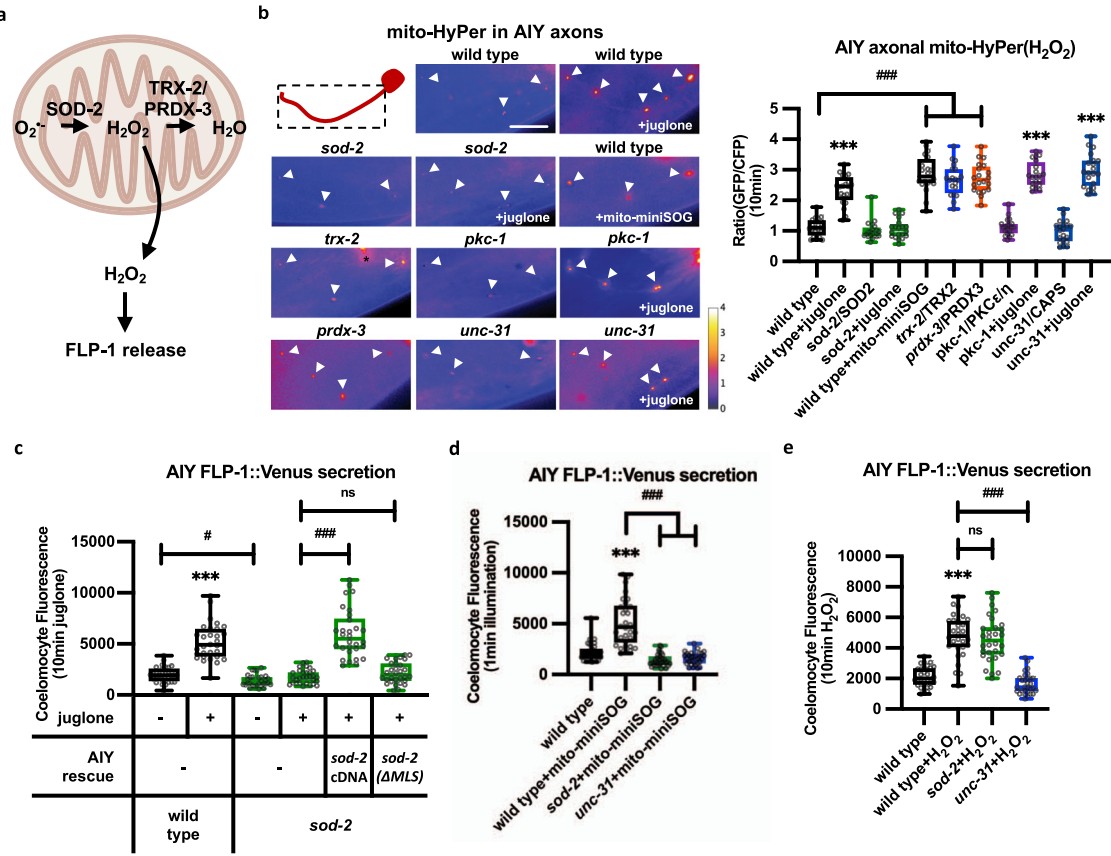

**Fig. 4 FLP-1 secretion from AIY interneurons relies on mitochondria-generated H$_2$O$_2$. a** Schematic representation of the generation of mitochondrial hydrogen peroxide (mtH$_2$O$_2$) by dismutation of superoxide (O$_2^-$) by mitochondrial SOD-2/superoxide dismutase, and its consumption by the mitochondrial TRX-2/thioredoxin PRDX-3/peroxiredoxin system, which converts mtH$_2$O$_2$ to H$_2$O. **b** Representative images and quantification of fluorescence in AIY axons of the indicated transgenic animals expressing mitochondrial-targeted HyPer (mito-HyPer) driven by *ttx-3* promoter. Arrowheads indicate mitochondrial fluorescence in AIY axons. The 520 nm/480 nm (GFP/CFP) ratio of HyPer was used to measure mtH$_2$O$_2$ levels. The boxes span the interquartile range, median is marked by the line and whiskers indicate the minimum and the maximum values. $n = 20$ biologically independent samples. ***$P < 0.001$ by Student's two-tailed *t*-test. ###$P < 0.001$ by one-way ANOVA with Dunnett's test. Scale bar: 10 μm. **c** Quantification of average coelomocyte fluorescence of the indicated mutants expressing FLP-1::Venus fusion proteins in AIY following vehicle (DMSO) or juglone treatment for 10 min. AIY rescue denotes expression of the indicated *sod-2* cDNAs under control of the *ttx-3* promoter. *sod-2*(ΔMLS) denotes *sod-2* cDNA in which the mitochondrial localization sequence has been removed. The boxes span the interquartile range, median is marked by the line and whiskers indicate the minimum and the maximum values. $n = 30$ biologically independent samples. ***$P < 0.001$ by Student's two-tailed *t*-test. #$P < 0.05$, ###$P < 0.001$, n.s not significant by one-way ANOVA with Dunnett's test. **d** Quantification of average coelomocyte fluorescence of the indicated mutants co-expressing FLP-1::Venus and mito-miniSOG in AIY following 1 min illumination with 50 mW/cm$^2$ blue light and 10 min recovery. The boxes span the interquartile range, median is marked by the line and whiskers indicate the minimum and the maximum values. $n = 30$ biologically independent samples. ***$P < 0.001$ by Student's two-tailed *t*-test. ###$P < 0.001$ by one-way ANOVA with Dunnett's test. **e** Quantification of average coelomocyte fluorescence of the indicated mutants expressing FLP-1::Venus fusion proteins in AIY following 10 min vehicle (M9) or H$_2$O$_2$ treatment. H$_2$O$_2$ treatment increased FLP-1 secretion in wild-type, *sod-2* and *ric-7* mutants, but not in *unc-31* mutants. The boxes span the interquartile range, median is marked by the line and whiskers indicate the minimum and the maximum values. $n = 30$ biologically independent samples. ***$P < 0.001$ by Student's two-tailed *t*-test. ###$P < 0.001$, n.s not significant by one-way ANOVA with Dunnett's test.

increases in mitochondrial H$_2$O$_2$ levels in AIY (Fig. 4b). The rapid coelomocyte fluorescence increase caused by juglone treatment eliminated the possibility that it arose through increased *flp-1::Venus* transgene expression or protein synthesis. Juglone treatment did not impact constitutive secretion of signal sequence-tagged Venus (P*ttx-3::ss-Venus*) from AIY, (Supplementary Fig. 3h[93]), suggesting that juglone does not generally boost secretion, nor does it alter bulk endocytosis of coelomocytes. We conclude that juglone-induced ROS production leads to a rapid increase in FLP-1 secretion from DCVs in AIY.

To address the specificity of juglone on DCV secretion, we first examined secretion of FLP-18, which is a FMRF-like neuropeptide protein whose release from AIY regulates foraging behavior and fat metabolism[42]. Unlike *flp-1* mutants, *flp-18* mutants were

not sensitive to juglone toxicity (Supplementary Fig. 3i). FLP-18::mCherry fusion proteins adopted a punctate pattern of localization in AIY axons that was similar to that of FLP-1::Venus (Supplementary Fig. 3j), and FLP-18::Venus secretion from AIY was dependent upon *unc-31*/CAPS (Supplementary Fig. 3k). However, FLP-18::Venus secretion from AIY was not significantly altered by juglone treatment (Supplementary Fig. 3k). Second, we examined FLP-1 secretion from the AVK interneuron, which secretes FLP-1 to regulate locomotion[68]. Animals expressing FLP-1::Venus selectively in AVK (under the P*flp-1*(513 bp) promoter[68]), exhibited a punctate pattern of fluorescence along AVK axons, and FLP-1::Venus secretion from AVK was dependent upon *unc-31*/CAPS and increased after food withdrawal (Supplementary Fig. 3l–n), as reported. FLP-1::Venus

secretion from AVK was not significantly different in animals treated with juglone compared to untreated controls (Supplementary Fig. 3m). Finally, we examined secretion of the insulin-like protein INS-22::Venus, which is released from motor neurons in an *unc-31*/CAPS-dependent manner[78]. Juglone treatment had no detectable effect on coelomocyte fluorescence intensity of INS-22::Venus-expressing animals (Supplementary Fig. 3o). Together these results indicate that the effect of juglone on DCV secretion is dictated by cellular context as well as the identity of DCV cargo, and that juglone may exhibit specificity in promoting FLP-1 secretion from AIY.

We next considered whether ROS production cell-autonomously in AIY could promote FLP-1 secretion. To test this, we activated mito-miniSOG with blue light for short times, which elicits ROS production without cell death[94]. As expected, a 1 min light exposure of animals expressing mito-miniSOG selectively in AIY resulted in a significant elevation of mitochondrial $H_2O_2$ levels in AIY 10 min later (Fig. 4b), without altering mitochondrial mass (Supplementary Fig. 3b). We detected a corresponding ~2-fold increase in FLP-1::Venus secretion following mito-miniSOG activation, similar to the increase caused by juglone or $H_2O_2$ (Fig. 4d). Mito-miniSOG activation failed to induce FLP-1 secretion in the absence of *sod-2* or *unc-31* (Fig. 4d). These results indicate that ROS produced by mitochondria in AIY itself can promote FLP-1 secretion.

To determine whether $H_2O_2$ can elicit FLP-1 secretion, we treated animals with $H_2O_2$, which has been shown to reach micromolar levels in *C. elegans* tissues within minutes following exposure[90]. We found that $H_2O_2$ treatment for 10 min robustly induced FLP-1 secretion to a similar extent as juglone treatment, and the increase was completely blocked by *unc-31* mutations (Fig. 4e). Unlike juglone treatment, $H_2O_2$ increased FLP-1 secretion in the absence of *sod-2* (Fig. 4e). Thus, exogenous $H_2O_2$ can bypass the requirement for mitochondrial $H_2O_2$ production to rapidly induce FLP-1 secretion from AIY.

**Mitochondrial calcium influx is required for ROS-induced FLP-1 secretion.** Mitochondrial calcium uptake regulates neurotransmitter and neuropeptide secretion in a variety of systems[3,4,95]. We found that juglone treatment significantly increased mitochondrial calcium levels (as measured by mito-GCaMP3) in wild-type animals, *sod-2*/SOD2 mutants, and *unc-2*/VGCC mutants (Fig. 5a). *mcu-1* encodes the *C. elegans* ortholog of the mitochondrial calcium uniporter[96], and *mcu-1* mediates uptake of calcium and ROS production in mitochondria of *C. elegans* skin in response to injury[97]. In the absence of juglone, *mcu-1* null mutants had normal mitochondrial calcium levels (Fig. 5a), $mtH_2O_2$ levels (Fig. 5b) and FLP-1::Venus secretion (Fig. 5C). However, juglone treatment failed to increase mitochondrial calcium levels, $mtH_2O_2$ levels, or FLP-1::Venus secretion in *mcu-1* mutants. The FLP-1::Venus secretion defects of *mcu-1* mutants were fully rescued by expressing *mcu-1* cDNA selectively in AIY (Fig. 5d). *micu-1* encodes the ortholog of MICU1 (mitochondrial calcium uptake 1), a regulatory subunit of MCU-1 that is proposed to promote the retention of accumulated calcium inside the mitochondrial matrix[98,99]. As expected, disruption of *micu-1* blocked juglone-induced FLP-1::Venus secretion (Supplementary Fig. 4a). *vdac-1* encodes the voltage dependent anion channel that transports ATP and other small metabolites across the outer mitochondrial membrane[100]. *vdac-1* disruption did not significantly alter juglone-induced FLP-1::Venus secretion (Supplementary Fig. 4a). These results suggest that mitochondrial calcium entry in AIY is critical for juglone-induced increases in $mtH_2O_2$ levels and FLP-1 secretion.

**TRX-2/thioredoxin and PRDX-3/peroxiredoxin regulate AIY $mtH_2O_2$ levels and FLP-1 secretion.** To determine whether $H_2O_2$ produced by mitochondria under normal physiological conditions contributes to FLP-1 secretion, we sought to genetically increase $mtH_2O_2$ levels in AIY without experimentally inducing mitochondrial stress. The peroxiredoxin-thioredoxin system is an evolutionarily conserved antioxidant system that specifically removes $H_2O_2$, thereby antagonizing $H_2O_2$ action. Peroxiredoxin converts $H_2O_2$ to water by catalyzing the transfer of oxidizing equivalents from $H_2O_2$ to a reactive cysteine residue at the peroxiredoxin active site. Thioredoxins, in turn, reduce oxidized peroxiredoxins for reuse to consume additional $H_2O_2$ molecules[101,102]. Once oxidized, thioredoxins themselves are reduced and thereby recycled by thioredoxin reductases (TrxR), utilizing NADPH as a cofactor (Fig. 5e[103]). *C. elegans* encodes one ortholog each of mitochondrial peroxiredoxin, PRDX-3[104], mitochondrial thioredoxin, TRX-2[105], and mitochondrial thioredoxin reductase, TRXR-2[105,106]. Null mutants in *prdx-3*, *trx-2*, or *trxr-2* exhibited significantly increased FLP-1::Venus secretion compared to wild-type controls (Fig. 5f, g, Supplementary Fig. 4c), whereas mutants of the cytoplasmic peroxiredoxin, *prdx-2* exhibited wild-type FLP-1 secretion (Fig. 5g, Supplementary Fig. 4c). The increase in FLP-1 secretion of *trx-2* mutants was blocked by *sod-2* mutations but was not further increased by juglone treatment (Fig. 5g). *trx-2* or *prdx-3* mutants also exhibited increased mito-HyPer punctal fluorescence intensity in AIY compared to wild-type controls (Fig. 4b). These results suggest that the peroxiredoxin-thioredoxin system negatively regulates FLP-1 secretion from AIY by reducing $mtH_2O_2$ levels.

*trx-2* is expressed in a small number of neurons and is highly expressed in AIY[105]. Expression of full length *trx-2* cDNA selectively in AIY fully restored wild-type FLP-1::Venus secretion to *trx-2* mutants (Fig. 5g). TRX-2 contains an MLS on its N terminus, and a conserved thiol-disulfide active site that contains reactive cysteines used for peroxiredoxin reduction (Supplementary Fig. 4d). Deletion of the TRX-2 MLS disrupted the mitochondrial localization of TRX-2::GFP fusion proteins in AIY axons (Supplementary Fig. 4e), and *trx-2*(ΔMLS) transgenes failed to rescue the increased FLP-1::Venus secretion defects of *trx-2* mutants (Fig. 5g). Similarly, *trx-2* transgenes with mutations in the active site predicted to impair catalytic activity (TRX-2 (ΔCAT)[105]) failed to rescue the FLP-1::Venus secretion defects of *trx-2* mutants (Fig. 5g). These results reveal that TRX-2 functions in AIY mitochondria to catalyze the removal of $mtH_2O_2$ and inhibit FLP-1 secretion, and they point to a critical role of endogenously produced $mtH_2O_2$ in AIY as a signaling cue that positively regulates FLP-1 secretion.

**Localized $H_2O_2$ production by axonal mitochondria regulates FLP-1 secretion.** Since $H_2O_2$ is readily neutralized by cellular antioxidant mechanisms in the cytosol, we hypothesized that if $mtH_2O_2$ promotes FLP-1 secretion, then mitochondria should be positioned in close proximity to DCV release sites in AIY axons. Consistent with this, we found that TOMM-20::mCherry puncta overlapped with FLP-1::Venus puncta along the entire length of AIY axons (Fig. 6a). To determine whether proximity of mitochondria and DCVs is important for FLP-1 secretion, we examined mutants in which axonal mitochondria and DCVs had been genetically separated. *ric-7* encodes an adapter protein that is selectively required for the anterograde transport of mitochondria to axons from the soma[107]. *ric-7* mutations nearly eliminated axonal TOMM-20::mCherry puncta without detectibly altering axonal FLP-1::Venus puncta, implying a specific disruption of mitochondrial but not DCV trafficking in AIY axons (Fig. 6a).

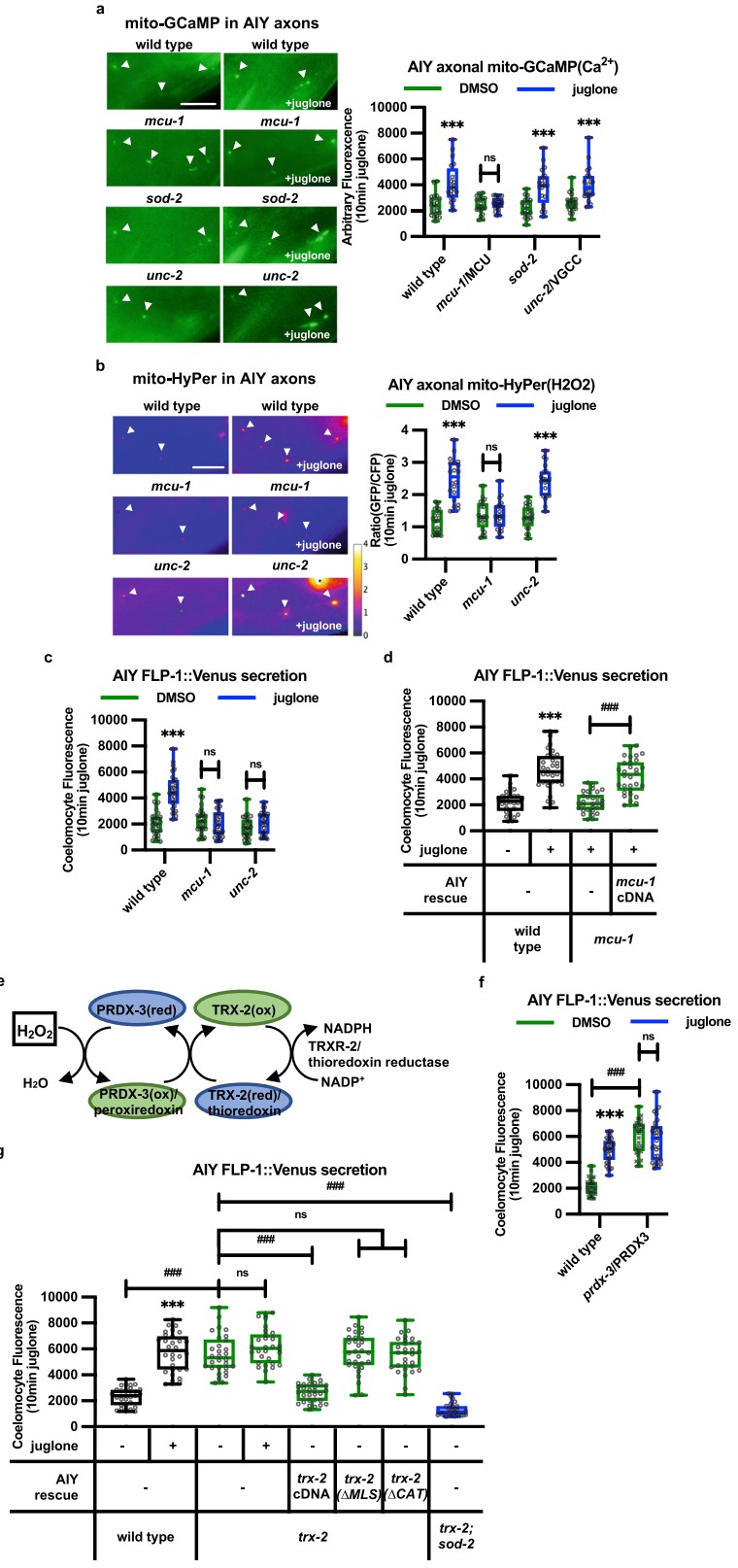

*ric-7* mutations significantly reduced baseline FLP-1::Venus secretion from AIY compared to wild-type controls, consistent with its reported role in positively regulating DCV secretion[108]. In addition, *ric-7* mutations abolished juglone-induced FLP-1:: Venus secretion (Fig. 6b). Expression of a chimeric protein (UNC-116::TOM-7, aka mito-truck), which functions as an adapter between mitochondria and microtubules that can restore

mitochondrial trafficking to *ric-7* mutants (Supplementary Fig. 5[86,107]), rescued both the mitochondrial trafficking defects and juglone-induced FLP-1::Venus secretion defects of *ric-7* mutants (Fig. 6a, b).

*unc-104*/KIF1A encodes a kinesin motor protein that mediates anterograde transport of DCVs from the soma, where they are generated, into axons[109], and *unc-104* is required for DCV

**Fig. 5 Regulation of FLP-1 secretion by mitochondrial calcium and the thioredoxin-peroxiredoxin system. a** Representative images and punctal fluorescence quantification of AIY axons from adults expressing mitochondrial-targeted GCaMP3 (mito-GCaMP) driven by the *ttx-3* promoter after 10 min vehicle (DMSO) or juglone treatment. The boxes span the interquartile range, median is marked by the line and whiskers indicate the minimum and the maximum values. $n = 20$ biologically independent samples. ***$P < 0.001$, n.s not significant by Student's two-tailed *t*-test. Scale bar: 10 μm. **b** Representative images and punctal fluorescence quantification of AIY axons from adults expressing mitochondrial-targeted HyPer (mito-HyPer) driven by the *ttx-3* promoter after 10 min vehicle (DMSO) or juglone treatment. The 520 nm/480 nm (GFP/CFP) ratio of HyPer was used to measure $H_2O_2$ levels. The boxes span the interquartile range, median is marked by the line and whiskers indicate the minimum and the maximum values. $n = 20$ biologically independent samples. ***$P < 0.001$, n.s not significant by Student's two-tailed *t*-test. Scale bar: 10 μm. **c** Quantification of average coelomocyte fluorescence intensity following exposure of the indicated mutants expressing FLP-1::Venus in AIY to juglone for 10 min. The boxes span the interquartile range, median is marked by the line and whiskers indicate the minimum and the maximum values. $n = 30$ biologically independent samples. ***$P < 0.001$, n.s not significant by Student's two-tailed *t*-test. **d** Quantification of average coelomocyte fluorescence intensity following exposure of *mcu-1* mutants expressing FLP-1::Venus in AIY to juglone for 10 min. AIY rescue denotes a transgene expressing *mcu-1* cDNA under the *ttx-3* promoter. The boxes span the interquartile range, median is marked by the line and whiskers indicate the minimum and the maximum values. $n = 30$ biologically independent samples. $n = 30$ biologically independent samples. ***$P < 0.001$, ###$P < 0.001$ by Student's two-tailed *t*-test. **e** Schematic showing the redox cycle used by peroxiredoxin, thioredoxin, and thioredoxin reductase to consume $H_2O_2$. **f** Quantification of average coelomocyte fluorescence intensity following exposure of *prdx-3*/peroxiredoxin3 mutants expressing FLP-1::Venus in AIY to juglone for 10 min. The boxes span the interquartile range, median is marked by the line and whiskers indicate the minimum and the maximum values. $n = 30$ biologically independent samples. ***$P < 0.001$, n.s not significant by Student's two-tailed *t*-test. ###$P < 0.001$ by one-way ANOVA with Dunnett's test. **g** Quantification of average coelomocyte fluorescence intensity following exposure of *trx-2*/thioredoxin2 mutants expressing FLP-1::Venus in AIY to juglone for 10 min. AIY rescue denotes transgenes that express wild-type, mitochondrial localization signal-deleted (*ΔMLS*), or catalytically inactive (*ΔCAT*) *trx-2* cDNA, under the *ttx-3* promoter. The boxes span the interquartile range, median is marked by the line and whiskers indicate the minimum and the maximum values. $n = 30$ biologically independent samples. ***$P < 0.001$ by Student's two-tailed *t*-test. ###$P < 0.001$, n.s not significant by one-way ANOVA with Dunnett's test.

secretion[110]. *unc-104* mutations reduced axonal FLP-1::Venus puncta without detectibly altering TOMM-20::mCherry puncta, indicative of a specific disruption of DCV trafficking from the soma (Fig. 6a). We found that *unc-104* mutations significantly reduced baseline FLP-1::Venus secretion from AIY compared to wild-type controls, and abolished juglone-induced FLP-1::Venus secretion. Double mutants lacking both *ric-7* and *unc-104*, which were largely devoid of both axonal mitochondria and DCVs (Fig. 6a), had defects in baseline and juglone-induced FLP-1:: Venus secretion that were no more severe than single mutants (Fig. 6b). Together, these results suggest that the proximity of mitochondria and FLP-1-containing DCVs in axons is critical for juglone-induced FLP-1 secretion. Importantly, we found that the increase in FLP-1::Venus secretion caused by mito-miniSOG activation was blocked by *ric-7* mutations (Fig. 4d). However, *ric-7* mutations failed to block $H_2O_2$-induced FLP-1::Venus secretion (Fig. 6c), indicating that exogenous $H_2O_2$ can bypass the need for axonal mitochondria. These results further support the idea that mt$H_2O_2$ generated locally at DCV release sites in axons is essential for FLP-1 release.

**PKC-1/PKC mediates the effects of $H_2O_2$ on FLP-1 secretion.** What are the molecular mechanisms by which $H_2O_2$ promotes FLP-1 secretion and the antioxidant response? $H_2O_2$ modifies reactive cysteine residues on target proteins, converting the thiol groups (–SH) into sulfenic acid (–SOH), in a process known as sulfenylation (Fig. 7a). Sulfenylation is reversible and typically changes the conformation and/or activity of proteins. We hypothesized that $H_2O_2$ might sulfenylate a target protein that regulates DCV exocytosis. PKCs are a family of serine/threonine kinases that are implicated in the regulation of secretory vesicle exocytosis and synaptic plasticity[111–117], and PKC family members are important intracellular targets for the effects of $H_2O_2$ on neuronal function[11,118–120]. *pkc-1*/PKCε/η encodes the sole member of the calcium-independent PKC subfamily. *pkc-1* positively regulates neuro-secretion from a number of neuron subtypes[78,121–123], and *pkc-1* has been implicated in mtROS signaling[124,125]. We found that *pkc-1* null mutations significantly reduced FLP-1::Venus secretion from AIY (Fig. 7b) without altering FLP-18::Venus secretion (Supplementary Fig. 3k). *pkc-1* mutations blocked the increased FLP-1::Venus secretion induced

by juglone, $H_2O_2$, mito-miniSOG activation, or *trx-2* mutation (Fig. 7b). The block in $H_2O_2$-induced FLP-1 secretion by *pkc-1* mutations cannot be explained by impaired mt$H_2O_2$ production (or transgene expression) because *pkc-1* mutants exhibited similar baseline and juglone-induced increases in AIY mito-HyPer fluorescence intensity as wild-type controls (Fig. 4b). Expressing *pkc-1* cDNA selectively in AIY fully restored juglone-induced FLP-1::Venus secretion to *pkc-1* mutants (Fig. 7c).

*pkc-1* mutants were hypersensitive to toxicity caused by juglone (Fig. 1a), and exhibited normal baseline SKN-1 activity, but had significantly decreased juglone-induced SKN-1::GFP nuclear translocation and P*gst-4::gfp* expression compared to wild-type controls (Fig. 7d, e). Thus, PKC-1 functions cell-autonomously in AIY to selectively promote mt$H_2O_2$-induced FLP-1 secretion and subsequent SKN-1 activation in the intestine.

**PKC-1(C524) is necessary for juglone-induced FLP-1 secretion.** PKC-1 contains a putative redox active cysteine residue (C524) in a conserved motif that shares striking similarity to the redox-sensitive region of IRE1, a protein kinase whose sulfenylation by $H_2O_2$ on the corresponding reactive cysteine regulates responses to ER stress (Fig. 7f[46]). C524 is conserved in mammalian calcium-independent PKC, and is located adjacent to a conserved basic amino acid predicted to stabilize the sulfenyl modification[126], and the DFG motif, which lies within the activation loop of the PKC-1 kinase domain and is critical for kinase activity[127]. Biochemical studies have shown that mammalian PKCα becomes robustly sulfenylated by $H_2O_2$ in cell culture[46,128]. To test whether $H_2O_2$-induced FLP-1 secretion relies on PKC-1 sulfenylation, we substituted C524 with serine, which replaces the thiol group (–SH) with a hydroxyl group (–OH) rendering it unable to be sulfenylated by $H_2O_2$ (Fig. 7a, f[46]). The C524S substitution did not appear to alter PKC-1 stability, distribution, or kinase activity since PKC-1(C524S)::GFP fusion proteins adopted similar fluorescence intensities and localization patterns in AIY axons as PKC-1(+):: GFP controls (Supplementary Fig. 6a), and *pkc-1*(C524S) transgenes fully rescued the *pkc-1* mutant's FLP-1::Venus secretion defects when expressed in AIY, and the INS-22::Venus secretion defects when expressed in motor neurons (under the *unc-129* promoter) in the absence of juglone (Fig. 7f and Supplementary Fig. 6B). However, *pkc-1*(C524S) transgenes failed to restore

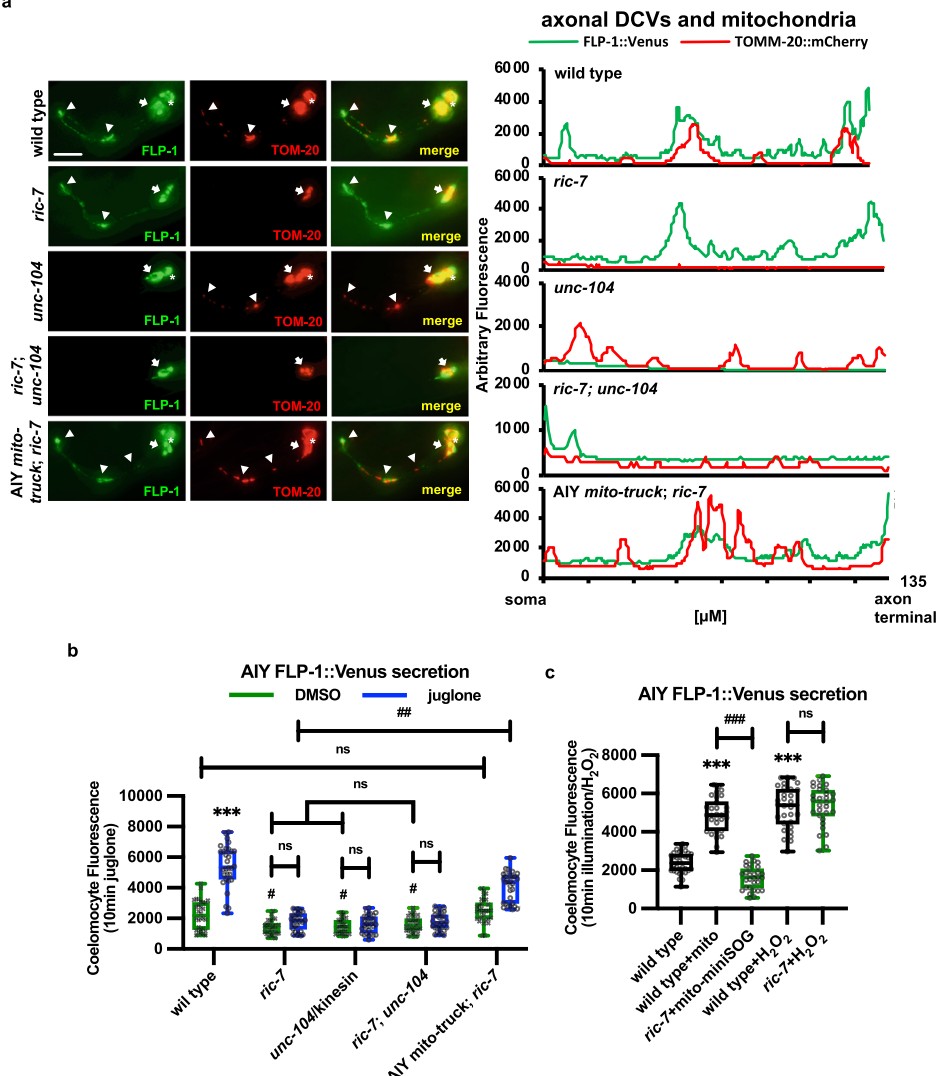

**Fig. 6 Trafficking of mitochondria and DCVs to AIY axons is required for juglone-induced FLP-1 secretion. a** Representative images and linescans of fluorescence distribution in AIY axons of animals co-expressing FLP-1::Venus (to mark DCVs) and TOMM-20::mCherry (to mark mitochondria) in the indicated mutants. Arrowheads mark fluorescent puncta in AIY axons and arrows mark AIY somas. Scale bar: 10 μm. **b** Quantification of average coelomocyte fluorescence intensity of the indicated mutants expressing FLP-1::Venus in AIY following 10 min juglone treatment. The boxes span the interquartile range, median is marked by the line and whiskers indicate the minimum and the maximum values. $n = 30$ biologically independent samples. ***$P < 0.001$ by Student's two-tailed $t$-test. ##$P < 0.01$, n.s not significant by one-way ANOVA with Dunnett's test. **c** Quantification of average coelomocyte fluorescence intensity of the indicated mutants expressing FLP-1::Venus in AIY following 10 min $H_2O_2$ treatment or 1 min mito-miniSOG activation. The boxes span the interquartile range, median is marked by the line and whiskers indicate the minimum and the maximum values. $n = 30$ biologically independent samples. ***$P < 0.001$ by Student's two-tailed $t$-test. ###$P < 0.001$, n.s not significant by one-way ANOVA with Dunnett's test.

juglone-induced FLP-1::Venus secretion from AIY to $pkc$-$1$ mutants (Fig. 7f). These results suggest that sulfenylation of C524 of PKC-1 plays a specific and critical role in mediating the effects of $H_2O_2$ on FLP-1 secretion without altering PKC-1 kinase activity.

Further oxidation by $H_2O_2$ converts sulfenic acid into sulfinic acid ($-SO_2H$) and subsequently to sulfonic acid ($-SO_3H$) (Fig. 7a[129]). To test whether further oxidation of C524 can promote secretion, we next generated the PKC-1(C524D) substitution, which is predicted to mimic sulfonylated cysteine (Fig. 7a, f). PKC-1(C524D) variants were expressed at similar levels and distribution in AIY as PKC-1(+) controls (Supplementary Fig. 7a), and rescued the baseline FLP-1::Venus secretion of $pkc$-$1$ mutants, but failed to restore juglone-induced FLP-1 secretion to $pkc$-$1$ mutants (Fig. 7f). This suggests that sulfonic acid modified C524 is not likely to contribute to FLP-1::Venus

secretion in response to stress. We conclude that $H_2O_2$-mediated sulfenylation or sulfinylation but not sulfonylation of PKC-1 C524 promotes FLP-1 secretion.

**Bacterial food sources alter $H_2O_2$ levels, FLP-1 secretion, and oxidative stress response.** Finally, we investigated whether the antioxidant response activated by FLP-1 signaling is regulated by environmental cues. AIY receives direct synaptic input from several olfactory and gustatory sensory neurons[130]. Therefore, we tested whether sensory input regulates FLP-1 secretion from AIY. $osm$-$6$ encodes a homolog of mammalian intraflagellar transport 52 protein that is necessary for cilium biogenesis in sensory neurons that provide synaptic input to AIY, and $osm$-$6$ mutants have defects in both olfactory and gustatory chemosensation[131,132]. We found that $osm$-$6$ mutants did not

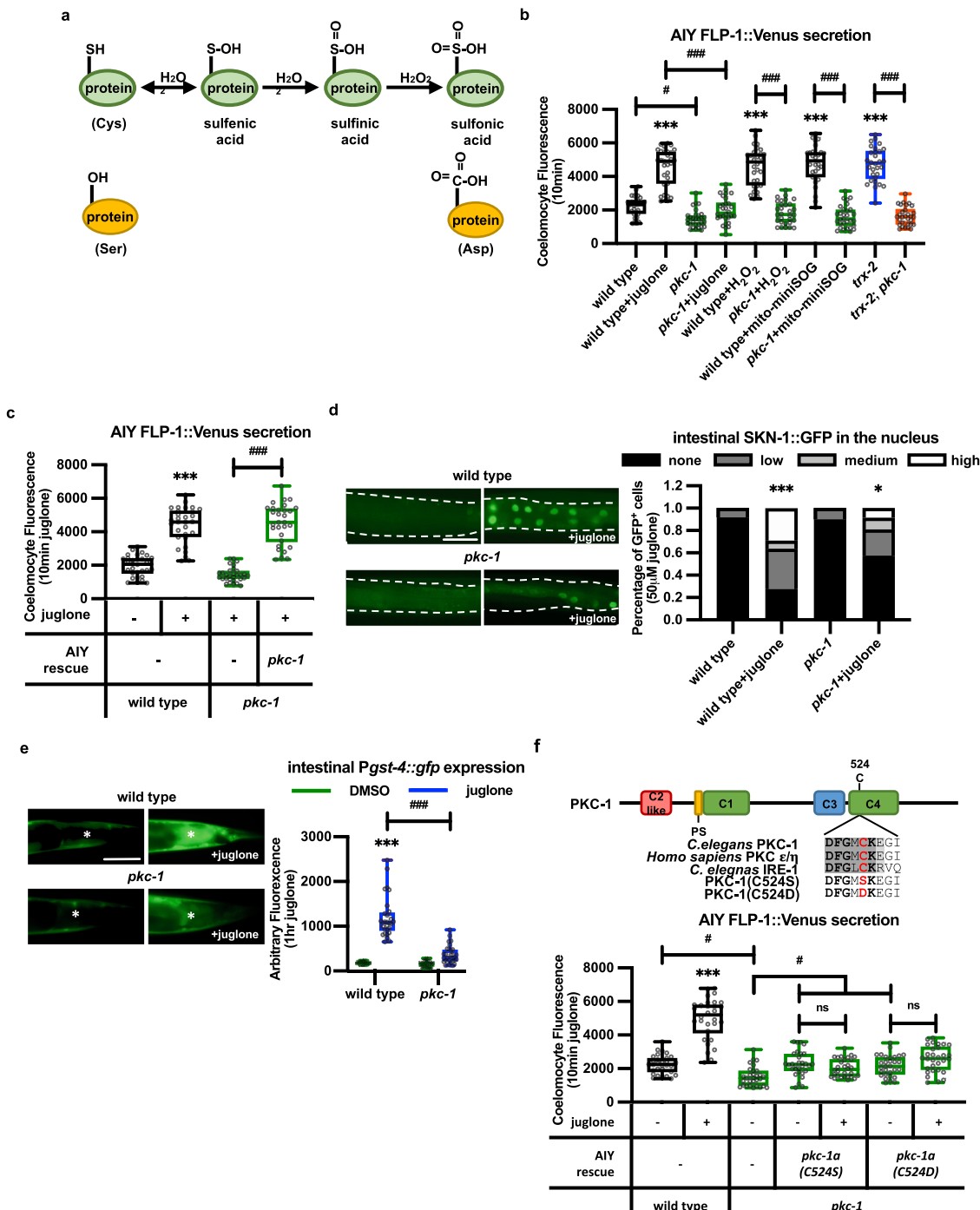

exhibit altered FLP-1 secretion either in the absence or presence of juglone (Supplementary Fig. 7a). Thus, the $H_2O_2$-induced FLP-1 secretion is unlikely to require synaptic input to AIY from the sensory system.

*C. elegans* is an obligate bacterial eater, and the types and abundance of bacteria that *C. elegans* consumes has dramatic effects on mitochondrial function and metabolism. We found that food restriction does not impact FLP-1 secretion because mutations in *eat-2*, which cause decreased feeding rates and are a classic model for caloric restriction[43,133,134], had no effect on baseline or juglone-induced coelomocyte fluorescence in animals expressing FLP-1::Venus (Supplementary Fig. 8a). Similarly, we found that food deprivation by starvation, which increased FLP-1

secretion from AVK (Supplementary Fig. 4m), did not alter FLP-1::Venus secretion from AIY (Supplementary Fig. 7b).

To determine whether differences in food source impact FLP-1 secretion we compared animals reared on the standard *Escherichia coli* OP50 strain (which was used throughout this study) to animals reared on HT115. *C. elegans* reared on OP50 have a number of differences in mitochondrial function and metabolism compared to those reared on HT115. Notably, animals grown on OP50 exhibit greater sensitivity to killing by oxidants, including $H_2O_2$ and juglone compared to animals grown on HT115[135], suggesting a reduced antioxidant capacity of OP50-fed animals. We found that animals reared on HT115 had similar mitochondrial distribution and abundance in AIY as

**Fig. 7 mtH$_2$O$_2$-induced FLP-1 secretion requires C524 of PKC-1. a** Schematic representation of progressive oxidation of the sulfhydryl groups (SH) of cysteine (Cys) residues by H$_2$O$_2$. The cysteine to serine (Ser) substitution mimics non-oxidizable cysteine, and the cysteine to aspartic acid (Asp) substitution mimics sulfonylated cysteine. **b** Quantification of average coelomocyte fluorescence intensity of the indicated mutants expressing FLP-1::Venus in AIY following 10 min treatment with the indicated oxidants. The boxes span the interquartile range, median is marked by the line and whiskers indicate the minimum and the maximum values. $n = 30$ biologically independent samples. ***$P < 0.001$ by Student's two-tailed $t$-test. #$P < 0.05$, ###$P < 0.001$ by one-way ANOVA with Dunnett's test. **c** Quantification of average coelomocyte fluorescence intensity of $pkc-1$ mutants expressing FLP-1::Venus in AIY following 10 min juglone treatment. AIY rescue denotes animals expressing $pkc-1a$ cDNA under control of the $ttx-3$ promoter. The boxes span the interquartile range, median is marked by the line and whiskers indicate the minimum and the maximum values. $n = 30$ biologically independent samples. ***$P < 0.001$ by Student's two-tailed $t$-test. ###$P < 0.001$ by one-way ANOVA with Dunnett's test. **d** Representative fluorescent images and quantification of the number of intestinal nuclei with SKN-1::GFP in adult animals before and after 10 min of juglone treatment. Dotted lines demarcate intestinal regions. Nuclear translocation of SKN-1::GFP was measured by counting the number of fluorescent nuclei in the intestine. Fewer than 10, between 11 and 20, and above 20 fluorescent nuclei are denoted Low, Medium, and High, respectively. $n = 36, 55, 49, 47$ biologically independent samples. *$P < 0.05$, ***$P < 0.001$ by Student's two-tailed $t$-test. Scale bar: 100 μm. **e** Representative images and quantification of the posterior regions of transgenic worms expressing the oxidative stress reporter P$gst-4::gfp$ after 1 h of juglone treatment and 4 h of recovery. Asterisks mark the intestinal region used for quantitative analysis. P$gst-4::gfp$ expression in body wall muscles, which appears as fluorescence on the edge of animals in some images, was not quantified. The boxes span the interquartile range, median is marked by the line and whiskers indicate the minimum and the maximum values. $n = 20, 29, 31, 31$ biologically independent samples. ***$P < 0.001$ by Student's two-tailed $t$-test. ###$P < 0.001$ by one-way ANOVA with Dunnett's test. Scale bar: 50 μm. **f** Top: Schematic representation of the protein structure of PKC-1, showing the conserved domains (C1-C4) the pseudosubstrate domain (PS) and the putative redox active region containing cysteine 524, which is conserved in human PKCε and $C. elegans$ IRE-1. Bottom: Quantification of average coelomocyte fluorescence intensity of the indicated mutants expressing FLP-1::Venus in AIY following 10 min treatment with juglone. AIY rescue denotes transgenes expressed under the $ttx-3$ promoter. The boxes span the interquartile range, median is marked by the line and whiskers indicate the minimum and the maximum values. $n = 30$ biologically independent samples. ***$P < 0.001$, n.s not significant by Student's two-tailed $t$-test. #$P < 0.05$ by one-way ANOVA with Dunnett's test.

animals reared on OP50 culture plates (Supplementary Fig. 7c). However, HT115-reared animals had 2-fold greater AIY mtH$_2$O$_2$ levels and 2.5-fold greater FLP-1::Venus secretion from AIY than animals grown on OP50 (Fig. 8a, b). The increase in FLP-1 secretion of HT115-reared animals was reduced to OP50 levels by $sod-2$ mutations (Fig. 8b). HT115-reared animals exhibited 3-fold greater intestinal P$gst-4::gfp$ expression compared to animals grown on OP50 (Fig. 8c). Heat-killed HT115 were as effective at increasing FLP-1 secretion as live HT115 (Fig. 8d), indicating that the signal that increases FLP-1 secretion is likely to be a bacterial metabolic product present in HT115 prior to its ingestion. These results indicate that the type of food source plays a critical role in setting the strength of both H$_2$O$_2$-regulated $flp-1$ signaling from AIY and antioxidant activity.

When animals grown on OP50 culture plates were switched to plates containing HT115, we observed a 2-fold increase in AIY H$_2$O$_2$ levels within 10 min and a 2.5-fold increase in FLP-1::Venus coelomocyte fluorescence within 30 min compared to non-switched animals (Fig. 8a, e). In the converse experiment, animals that were maintained on HT115 and switched to OP50 exhibited significantly reduced AIY mtH$_2$O$_2$ levels beginning about 60 min after being switched, with reductions in FLP-1::Venus coelomocyte fluorescence beginning about 120 min after the switch, and reaching levels seen in OP50-reared animals after 180 min (Fig. 8a, e). The lags between the changes in mtH$_2$O$_2$ levels and coelomocyte fluorescence likely reflect the time it takes for coelomocytes to uptake and degrade FLP-1::Venus following H$_2$O$_2$-regulated increases and decreases in FLP-1::Venus secretion, respectively. Together, these results indicate that the ingestion of (or exposure to) HT115 leads to the rapid and reversible increase in mtH$_2$O$_2$ in AIY and the subsequent increase in FLP-1 secretion. Importantly, the observation that changes in FLP-1 secretion are reversible within minutes upon switching food sources reinforces the idea that H$_2$O$_2$ performs a signaling function rather than functioning as a damaging oxidant, in which case changes would be expected to be slower or irreversible.

Animals grown on either OP50 or HT115 showed similar juglone-induced increases in FLP-1 secretion and intestinal P$gst-4::gfp$ expression (Fig. 8b, c), as well as increased sensitivity to juglone-mediated toxicity in the absence of $flp-1$ (Fig. 8f). However, OP50-reared animals exhibited significantly reduced

P$gst-4::gfp$ expression compared to HT115-reared animals following exposure to lower juglone concentrations that do not elicit maximal responses (Fig. 8c). These results suggest that OP50-reared animals have a less robust oxidative stress response than HT115 animals, likely due to lower baseline $flp-1$ signaling on this food source.

## Discussion

By analyzing the mechanisms by which the nervous system activates the antioxidant response, we have discovered a physiological role for H$_2$O$_2$ originating from mitochondria in regulating the secretion of a neuropeptide, FLP-1, that functions as a neuroendocrine stress signal to activate the oxidative stress response in distal tissues. We showed that FLP-1 secretion from the AIY interneurons is necessary and sufficient to promote activation of the antioxidant response in the intestine and organism-wide protection against oxidative stress. Exposure of animals to different types of bacterial food sources leads to rapid local changes in H$_2$O$_2$ levels in axonal mitochondria in AIY and to corresponding changes in FLP-1 secretion that depend on the mitochondrial superoxide dismutase, $sod-2$. We identified a role in AIY for the antioxidant peroxiredoxin-thioredoxin system in reducing mtH$_2$O$_2$ levels in axons and inhibiting FLP-1 secretion. The effects of H$_2$O$_2$ on FLP-1 secretion rely on a putative redox active cysteine residue in the serine threonine kinase, PKC-1.

We propose a function for AIY as a "stress sensing" neuron that responds to bacteria or other toxins that are ingested and signal to AIY from the alimentary system. When fed OP50 bacteria, mtH$_2$O$_2$ levels and $flp-1$ signaling in AIY are low. Upon changing food source to HT115, or exposure to oxidative stressors, levels of mtH$_2$O$_2$ are rapidly increased in axonal mitochondria in the vicinity of FLP-1 release sites. After exiting mitochondria, H$_2$O$_2$ functions as a signaling cue where it sulfenylates PKC-1 on C524, which in turn positively regulates exocytosis of FLP-1 containing DCVs. Once secreted, FLP-1 functions as a neuroendocrine stress signal that tunes the antioxidant response in distal tissues by activating NPR-4/GPCR and positively regulating SKN-1 signaling in the intestine. Under conditions when mtH$_2$O$_2$ levels are low in AIY or in the absence of $flp-1$, $pkc-1$, or $npr-4$, SKN-1 activation is compromised, and worms are more susceptible to the deleterious effects of oxidative

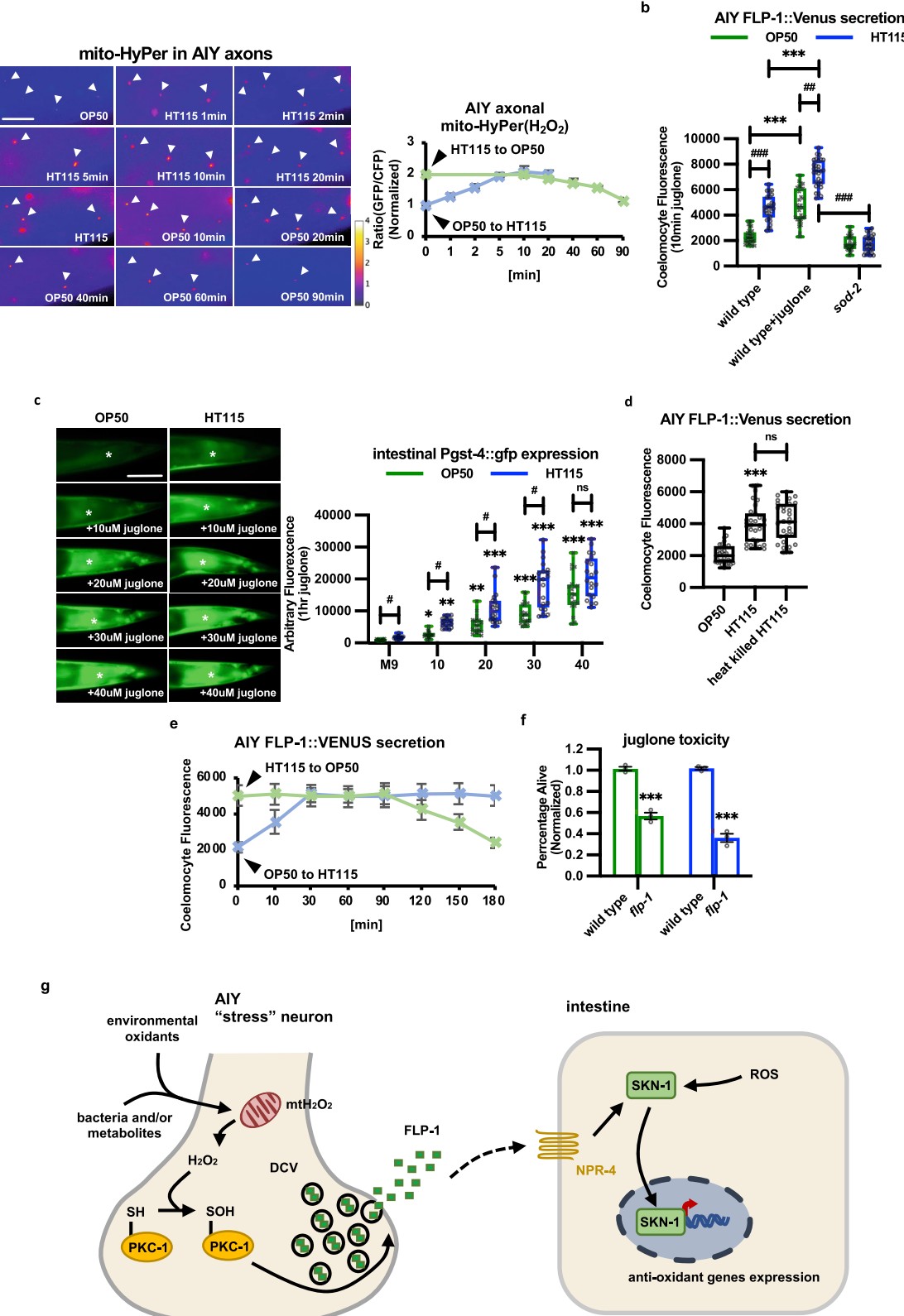

stress arising from environmental insults. Our data support a model whereby the regulated secretion of FLP-1 from AIY by mtH₂O₂ is a mechanism by which animals maintain antioxidant homeostasis as they adapt to changes in the composition of diverse food sources to ensure survival (Fig. 8g).

Neurons are among the most metabolically active cells, yet in mammals, mature neurons possess a limited capacity to neutralize ROS since Nrf2 activity is weak. Instead, neurons undergoing stress can take up antioxidant precursors that are secreted by neighboring astrocytes. The release of "stress signals" from neurons undergoing stress is proposed to lead to Nrf2 activation in astrocytes and the subsequent secretion of antioxidant precursors[49,136,137]. Although neuronal glutamate release has been suggested to activate Nrf2 in astrocytes, additional signals, possibly neuropeptides, are likely to

**Fig. 8 Bacterial food sources alter FLP-1 secretion and the antioxidant response. a** Representative images and punctal fluorescence quantification of AIY axons from adults expressing mitochondrial-targeted HyPer (mito-HyPer) driven by the *ttx-3* promoter cultured on OP50 or HT115 bacteria and switched to the other food source for the indicated times. The 520 nm/480 nm (GFP/CFP) ratio of HyPer was used to measure $H_2O_2$ production. Arrowheads mark fluorescent puncta in AIY axons. Data are mean values ± s.e.m. *n* = 20 biologically independent samples. Scale bar: 10 μm. **b** Quantification of average coelomocyte fluorescence intensity of animals expressing FLP-1::Venus in AIY cultured with either OP50 or HT115 bacteria with or without 10 min juglone treatment. The boxes span the interquartile range, median is marked by the line and whiskers indicate the minimum and the maximum values. *n* = 30 biologically independent samples. \*\*\**P* < 0.001 by Student's two-tailed *t*-test. ##*P* < 0.01, ###*P* < 0.001 by one-way ANOVA with Dunnett's test. **c** Representative images and quantification of average P*gst-4::gfp* expression in the posterior intestine of adult animals (asterisks) cultured on OP50 or HT115 following exposure to the indicated concentrations of juglone for 1 and 4 h recovery. P*gst-4::gfp* expression in body wall muscles, which appears as fluorescence on the edge of animals in some images, was not quantified. The boxes span the interquartile range, median is marked by the line and whiskers indicate the minimum and the maximum values. *n* = 20 biologically independent samples. \**P* < 0.05, \*\**P* < 0.01, \*\*\**P* < 0.001 by one-way ANOVA with Dunnett's test. #*P* < 0.05, n.s not significant by Student's two-tailed *t*-test. Scale bar: 50 μm. **d** Quantification of average coelomocyte fluorescence intensity of animals expressing FLP-1::Venus in AIY cultured on live or heat-killed HT115 bacteria. The boxes span the interquartile range, median is marked by the line and whiskers indicate the minimum and the maximum values. *n* = 30 biologically independent samples. \*\*\**P* < 0.001, n.s not significant by Student's two-tailed *t*-test. **e** Quantification of average coelomocyte fluorescence intensity of animals expressing FLP-1::Venus in AIY cultured on OP50 or HT115 bacteria and switched to the other food source for the indicated times. Data are mean values ± s.e.m. *n* = 30 biologically independent samples. **f** Percentage of surviving animals of the indicated genotypes 16 h following treatment of young adults with juglone for 4 h. Survival of *flp-1* mutants was normalized to wild-type animals raised on the same bacterial strain. Data are mean values ± s.e.m. *n* = 447, 426, 426, 423 biologically independent samples over three independent experiments. \*\*\**P* < 0.001 by Student's two-tailed *t*-test. **g** A schematic model in which cues originating from ingested bacteria (or other stressors) rapidly regulate mitochondrial $H_2O_2$ levels in AIY, leading to PKC-1 C524 sulfenylation and FLP-1 secretion from AIY axons. Once secreted, FLP-1 activates intestinal SKN-1 signaling and the antioxidant response through the GPCR NPR-4. The regulation of FLP-1 secretion by $H_2O_2$ may represent a way in which the nervous system can rapidly adjust antioxidant defenses in distal tissues in order to adapt to constantly changing food sources.

contribute to Nrf2 activation[136]. Interestingly, SKN-1 activation in the worm intestine not only activates the antioxidant response, but also leads to the secretion of diffusible signals that regulate neuronal function[138,139]. Thus, the activation of SKN-1/Nrf2 in distal tissues by neuronal activity, and the subsequent feedback signaling to neurons may be an evolutionary conserved mechanism by which neurons maintain redox homeostasis in response to oxidative stress.

Here we identified a previously undescribed function for the AIY interneuron pair in relaying changes in levels of mitochondrially generated $H_2O_2$ to the rest of the organism through FLP-1 secretion. The AIY pair is a first layer amphid interneuron that receives glutamatergic synaptic input from a number amphid sensory neurons to regulate food, odor, and thermal-evoked behaviors through cholinergic synaptic transmission[140]. AIY also plays a critical role in regulating development and metabolism in response to environmental signals through the secretion of FLP-1 and FLP-18[42,69,70]. Our observation that FLP-1 but not FLP-18 secretion from AIY is increased by juglone and decreased by *pkc-1* mutations suggests that FLP-1 and FLP-18 secretion may be differentially regulated, raising the possibility that these peptides are packaged into distinct DCV pools that may differ in their proximity to mitochondria, or their ability to be regulated by PKC-1 signaling. Consistent with this, differential release of DCV pools from single cells has been observed in a number of neuroendocrine cell types[141–143]. Thus, AIY appears to be unique among neurons we examined in its ability to specifically regulate FLP-1 secretion through $mtH_2O_2$ signaling, revealing a specialized function for AIY in the oxidative stress response. Interestingly, we found that *flp-1* mutants are sensitive to juglone-induced toxicity, whereas *flp-18* mutants are resistant to juglone-induced toxicity, and *flp-1; flp-18* double mutants exhibit an intermediate juglone response (Supplementary Fig. 3i), suggesting a role for *flp-18* in inhibiting the oxidative stress response by antagonizing the protective effects of *flp-1* signaling.

How might endogenous $H_2O_2$ levels in AIY be regulated to calibrate the strength of FLP-1 secretion? First, $H_2O_2$ levels could be regulated by controlling ROS production by mitochondria. Mitochondrial ROS production has been shown to be controlled by numerous signals including by glucose in pancreatic beta cells,

synaptic activity in neurons, and peptide hormones in smooth muscle cells[144–147]. In *C. elegans*, mitochondrial activity is also highly dynamic and can be influenced by intrinsic metabolic processes driven by changes in diet and the gut microbiota[135,148]. Second, $mtH_2O_2$ levels in AIY could be regulated by controlling the rate of $H_2O_2$ consumption by the peroxiredoxin-thioredoxin system[101]. In mammals, the activity of the mitochondrial peroxiredoxin-thioredoxin system can be regulated by cAMP arising from extracellular signals in the adrenal cortex[149], and by synaptic activity in cortical neurons[147]. Third, $H_2O_2$ levels could be regulated by controlling calcium influx into mitochondria. We found that juglone treatment leads to increased mitochondrial calcium levels, and that mitochondria calcium influx is required for $mtH_2O_2$ production and for juglone-induced mitochondrial calcium increases. Thus, juglone treatment may increase calcium levels in the cytosol that then enters the mitochondrial through MCU1. Alternatively, juglone may directly regulate calcium levels in the mitochondria for example by regulating MCU channel activity. The activity of the MCU channel is proposed to be modulated by a number of post-transcriptional modifications, including by phosphorylation, and by ROS mediated S-glutathionylation[150]. The mechanism by which increased mitochondrial calcium levels facilitate the generation of $H_2O_2$ is less clear. Calcium is an activator of the ETC, and sustained ETC activation can lead to mtROS production[151]. Finally, FLP-1 secretion could be controlled by regulating the rate of $mtH_2O_2$ exit from mitochondria possibly through aquaporins[152], or the extent of $H_2O_2$ buffering in the cytoplasm by cytosolic antioxidant enzymes[153], or the proximity of mitochondria to DCVs[86,154]. Regulation of any one or more of these steps by stress signals could modulate FLP-1 secretion from AIY.

Our studies reveal a central role for PKC-1/PKCε/η in mediating the effects of $H_2O_2$ on FLP-1 secretion. PKC has been implicated in a number of redox signaling events in excitable cells[155–157]. In hippocampal neuron cultures, PKC mediates the effects of ROS on long term plasticity[11]. In *Aplysia*, ROS activates novel PKC, which is important for the establishment of changes in synaptic strength[158]. PKCε protects neurons and cardiac myocytes in models of ischemia/reperfusion injury and is

activated by ROS produced by mitochondria during hypoxia[159]. In *C. elegans*, *pkc-1* is required for the protective effects of mitochondrial ROS generated by mutants with impaired mitochondrial respiration to an antimitotic toxin[124,160]. In addition, *pkc-1* has been shown to function in AIY to mediate the effects of the ROS generator graphene oxide on oxidative stress the intestine[125]. Whether *pkc-1* regulates FLP-1 secretion or the secretion of other signals during these responses remains to be determined.

Our genetic studies point to sulfenylation of the conserved PKC-1 C524 residue in promoting $H_2O_2$-induced FLP-1 secretion. Cysteine sulfenylation is a rapid enzyme-independent reversible modification[129], making this an attractive mechanism by which PKC-1 could rapidly and reversibly regulate FLP-1 secretion. $H_2O_2$ signaling can lead to cysteine sulfenylation either directly by oxidization of reactive cysteines on target proteins, or indirectly by oxidation of a "relay" protein such as peroxiredoxin, which then transfers oxidizing equivalents to the target protein[21]. We do not favor the idea that $H_2O_2$ leads to PKC-1 sulfenylation indirectly through PDRX-3, since we found that *prdx-3* mutations do not decrease FLP-1 release, which would be expected if PRDX-3 positively links $H_2O_2$ to PKC-1 activation. Instead, we propose that PKC-1 is either directly sulfenylated by $H_2O_2$ or is sulfenylated by an alternative redox-active "relay" protein.

TRX-2 is nearly exclusively expressed at high levels in AIY suggesting a unique capacity of AIY to remove $mtH_2O_2$ through the peroxiredoxin-thioredoxin system. How then could $H_2O_2$ in AIY reach levels high enough to promote FLP-1 secretion? One explanation, termed the "floodgate hypothesis", proposes that when $H_2O_2$ levels increase beyond a certain threshold, peroxiredoxins become hyperoxidized and are inactivated allowing $H_2O_2$ to build up sufficiently to perform its signaling functions[161,162]. In this scenario, PRDX-3 would remove $H_2O_2$ efficiently under normal conditions, but an increase in $mtH_2O_2$, arising from stress signals, would lead to PRDX-3 hyperoxidation resulting in further $mtH_2O_2$ accumulation to levels that would promote FLP-1 secretion. Interestingly, the function we found for *pdrx-3* in inhibiting $H_2O_2$ signaling contrasts with that of the cytoplasmic peroxiredoxin, *prdx-2*, which promotes $H_2O_2$ signaling by functioning as a redox-active "relay" protein for $H_2O_2$ in sensory neurons during sensory transduction[13,14], revealing distinct mechanisms by which cytoplasmic vs. mitochondrial peroxiredoxins impact $H_2O_2$ signaling in *C. elegans*.

Peroxiredoxins are efficient in $H_2O_2$ sensing and scavenging[101] and mammalian peroxiredoxin 3 is estimated to account for up to 90% of cellular $H_2O_2$ detoxification[163,164], suggesting that it may play a major role in mitochondrial redox signaling. Mouse peroxiredoxin 3 knockouts appear largely healthy, but have increased $mtH_2O_2$ levels, and have defects in mitochondrial and metabolic homeostasis in a number of tissues[165], whereas overexpression of peroxiredoxin 3 results in decreased ROS levels[166]. Peroxiredoxin 3 and thioredoxin 2 are expressed broadly in the nervous system, where they have well documented roles in neuroprotection in response to oxidative damage in hippocampal and cortical neurons[167–169]. Studies in insulinoma cells have shown that glucose-stimulated insulin release is enhanced upon peroxiredoxin 3 knockdown and blocked upon peroxiredoxin 3 overexpression[170], and peroxiredoxin 3 knockouts have elevated fasting insulin plasma levels and are insulin resistant[171], suggesting a potentially conserved function for *prdx-3* and mammalian peroxiredoxin 3 in inhibiting DCV secretion by the removal of $mtH_2O_2$. It will be interesting to determine whether the peroxiredoxin-thioredoxin system regulates neurotransmitter or neuropeptide secretion from mammalian neurons.

We found that cysteine sulfenylation of PKC-1 does not appear to affect kinase activity since the PKC-1(C524S) mutant transgenes

fully supported neuropeptide secretion in the absence of stress. PKCs are cytosolic proteins whose activities are primarily regulated by targeting to cellular membranes through their calcium and/or lipid binding C1 and C2 domains. In their role in promoting secretory granule secretion, the recruitment of PKC to release sites regulates a late step in vesicle exocytosis[172]. $H_2O_2$ treatment induces membrane translocation of PKCγ, which is involved in ischemia[128]. Therefore, we propose that $H_2O_2$-induced C524 sulfenylation promotes membrane recruitment of PKC-1 to DCV release sites, where it facilitates calcium-dependent FLP-1 release. PKC-1 may promote FLP-1 secretion through phosphorylation of known PKC target proteins involved in secretory granule exocytosis[115,173–176]. Alternatively, PKC-1 regulate mitochondrial function. Calcium-independent PKCs interact with a number of mitochondrial proteins[157], and have been proposed to promote insulin secretion by regulating mitochondrial ATP production in pancreatic beta cells[177,178]. Finally, PKC-1 may regulate DCV secretion through a less direct mechanism. For example, *pkc-1* is reported to regulate the expression of the MAP kinase cascade activator *lin-45*/Raf in AIY following exposure to oxidative stress[125].

Our results suggest that diet plays a critical role in setting the baseline antioxidant response through the regulation AIY $H_2O_2$ levels and FLP-1 secretion. OP50, which is a B strain and HT115, which is a K-12 strain, differ in both the amount and composition of nutrients and metabolites, including specific lipids, mitochondrial byproducts and amino acids[179,180], and some of these differences are thought to underly differences in mitochondrial function and stress responses in animals that consume each bacterial type. Because of the relatively fast rates at which changing diet leads to changes in FLP-1 secretion, we propose that a signal or metabolite from consumed bacteria is sensed either directly or indirectly by AIY, leading to changes in $H_2O_2$ levels in AIY. *C. elegans* also can ingest oxidants directly from food or from the environment. For example, some pathogenic bacteria consumed by *C. elegans* produce levels of $H_2O_2$ as high as 2 mM[181,182]. Similarly, juglone is a naturally occurring oxidant exuded into the soil by the roots of black walnut trees[183], where it may be consumed by *C. elegans* at physiologically relevant concentrations to impact ROS production in AIY. Determining how bacterial-derived signals from diet regulate $H_2O_2$-dependent neurosecretion from AIY may have general relevance for understanding mechanisms underlying the crosstalk between the nervous system and the gut microbiome.

## Methods

**Strains.** *C. elegans* strains were maintained on standard nematode growth medium (NGM) plates seeded with OP50 *E. coli* bacteria as food source, unless otherwise indicated, and cultured in a dark 20 °C incubator. All strains were synchronized by picking mid L4 stage animals and either analyzed immediately (for coelomocyte imaging) or as young adults 24 h later (for AIY imaging or toxicity assays). The wild-type reference strain was Bristol N2. Mutants used in this study were outcrossed at least two times and are listed in Supplementary Table 3.

Transgenic animals were generated by micro-injecting plasmid mixes into the gonads of young adult N2 hermaphrodites following standard techniques as previously described[184]. All microinjection mixes were prepared using expression constructs injected at 10 ng/μL, plus the co-injection markers KP#708 (Pttx-3-rfp, 40 ng/μL), KP#1106 (Pmyo-2-gfp, 10 ng/μL), pJQ70 (Pofm-1-rfp, 25 ng/μL) or pMH163 (Podr-1-mCherry, 40 ng/μL) to a final concentration of 100 ng/μL. At least three transgenic lines for each transgene were examined. Integration of arrays was performed by radiating 100 transgenic L4 worms on unseeded NGM plates with the lid removed in a UV cross-linker. 200 F1 transgenic worms were singled onto separate NGM plates, and F2 transgenic worms with high level of fluorescence were selected for candidates for 100% homozygous transgenic F3 worms[185]. Heat killing bacteria was performed by placing bacteria in a 100 °C water bath for ~30 min, verified by failure to form colonies on LB plates.

**Toxicity assays.** Stock solutions of 50 mM juglone in DMSO, 20 mM thimerosal in water, and 4 mM paraquat in water (each freshly made before each assay), 0.5% w/v sodium arsenite in water, and 30% (9.8 M) $H_2O_2$ solution were used for toxicity assays. High concentrations for treatment were required for some drugs

since the *C. elegans* cuticle is not permeable to most drugs[186]. For liquid toxicity assays, about 50–80 synchronized adults were transferred into a 1.5 mL Eppendorf tube with M9 buffer, and washed three times. Oxidants were added to washed worms at final concentrations of 100–300 μM (juglone), 50 μM (thimerosal), 2 mM (sodium arsenite), unless indicated otherwise, and incubated for 4 h on rotating mixer. Animals were then washed once and transferred to fresh plates seeded with OP50 or HT115 to recover in dark at 20 °C for 16 h. Survival was assayed by counting the number of alive and dead animals. For plate toxicity assays, plates containing 4 M fructose rings around the bacteria were freshly prepared to restrain the movement of worms. Toxicity assays were performed in triplicate.

**Cell ablation and ROS production by miniSOG.** For miniSOG-induced cell ablation, an EXFO mercury arc lamp was used as the blue light source. 30–40 L4 animals were transferred onto fresh NGM plates with OP50 and animals were exposed to continuous 50 mW/cM$^2$ blue light for 30 min and recovered at 20 °C in dark for 16 h before toxicity assays. Blue light illumination was performed in triplicate. For miniSOG-induced ROS generation, LED light with pre-built MSOG0001 filter module (TriTech Research) was used as the blue light source. 30–40 L4 stage worms were transferred onto fresh NGM plates with OP50 and animals were exposed to continuous 100 mW/cM$^2$ blue light for 10 min and recovered at 20 °C in dark for 10 min before taking images. Plates were placed without covers on the stage with blue light passing through without objectives.

**Channelrhodopsin activation.** Blue light-induced ChR2 activation was performed as previously described[82]. 100 mW/cm$^2$ blue LED light source was used for light illumination. Briefly, 30–40 animals expressing *ChR2::gfp* transgene were transferred onto NGM plates with OP50 spread supplemented with 500 μM all-*trans* retinal (Sigma), and recovered at 20 °C in dark for 4 h before light illumination. Plates were placed without covers on the stage with blue light passing through without objectives. For light illumination, animals were exposed to continuous blue light for 1 min and recovered at 20 °C in dark for 10 min before taking images.

**Microscopy and analysis.** Fluorescence microscopy experiments to quantify AIY axonal and coelomocyte fluorescence were performed as previously described[80]. Briefly, 30–40 animals were paralyzed with 30 mg/mL 2,3-butanedione monoxime (BDM) in M9 buffer and mounted on 2% agarose pads. Images were captured with the Nikon eclipse 90i microscope equipped with a Nikon PlanApo ×40 or ×100 objective (NA = 1.4) and a Hamamatsu Orca Flash LT + CMOS camera, and Metamorph 7.0 software (Universal Imaging/Molecular Devices) was used to capture serial image stacks and the maximum intensity projection image was used for analysis.

For transcriptional reporter imaging, 30–40 young adults were transferred into M9 solution, washed three times and then exposed to oxidant for 1 h on a rotating mixer. Animals were then washed three times and transferred onto fresh OP50-seeded NGM plates and allowed to recover in dark at 20 °C for 4 h before imaging. The posterior end of the intestine at the gonad bend was imaged using a ×60 objective for each reporter. For quantification of P*gst-4::gfp* expression, a 16-pixel diameter circle was drawn in the posterior intestine (ROI) and the average fluorescence intensity within the area of the circle was calculated using MetaMorph. Background fluorescence was measured as the average intensity within a same-sized circle positioned next to the animal (coverslip fluorescence) and this value was subtracted from the ROI fluorescence to generate the fluorescence intensity value. For SKN-1::GFP and DAF-16::GFP reporter imaging, 40–50 L4 animals were exposed to the working concentration of oxidant for 10 min in liquid, prior to imaging. Z-stacks of the intestine (20 nuclei/image) were captured using a ×40 objective and the total number of GFP$^+$ nuclei in the intestine were counted. Fluorescent nuclei between 1 and 10, 11 and 20, and above 20 were binned into Low, Medium and High categories, respectively.

For imaging fluorescently tagged fusion proteins in AIY, 30–40 synchronized young adults were exposed to either oxidant or M9 buffer for the indicated times (usually 10 min) and then paralyzed in BDM for 10 min prior to imaging. Only animals oriented such that the left AIY neuron was facing the objective were imaged. Z stacks of the axons were captured, and the fluorescence intensity values were then quantified by obtaining linescans in Metamorph and using custom Puncta 6.0 software written with Igor Pro (Wavemetrics) to illustrate fluorescence distribution or to quantify punctal fluorescence, as previously described (http://tock.bcgsc.ca/cgi-bin/ce2/eprofile)[80,187]. Z stacks were obtained using GFP (excitation/emission: 450 nm/520 nm) and CFP (excitation/emission: 420/480 nm) filter sets sequentially. mito-HyPer fluorescence amplitude was quantified as the ratio of GFP to CFP punctal fluorescence intensity changes with respect to the baseline [$(F_t − F_0)/F_0$].

For coelomocyte imaging, 30–40 L4 stage animals were washed three times with M9 and then exposed to oxidants for 10 min in M9 unless indicated otherwise. Animals were then paralyzed in BDM in M9 buffer before taking images of the posterior most coelomocytes at the posterior gonad bend. Z stacks were quantified by measuring average fluorescence/pixel of FLP-1::Venus from 2 to 4 endocytic in the posterior coelomocyte of each animal in Metamorph. All FLP-1::Venus secretion assays from AIY neurons were performed using the *vjIs150* transgene, which is integrated on LG III, except for *trxr-2* mutants, which were analyzed using *vjIs152*, which is integrated on LG I.

**RNA interference.** Feeding RNAi was performed as described[188]. Briefly, 20–25 gravid adult animals were placed on RNAi plates seeded with HT115(DE3) bacteria transformed with L4440 vector containing the insert of the gene targeted for knockdown or empty L4440 vector as a control, and eggs were collected for 4 h to obtain age-matched synchronized worm populations. Young adult animals were used for all RNAi assays. RNAi clones were from the Ahringer RNAi library[189], and clones not represented in the library were made from genomic DNA and confirmed by sequencing. See Supplementary Table 3 for a list of plasmids made.

**Statistical analysis.** Statistical analysis was performed on GraphPad Prism 8. Unpaired *t*-test (two tails) was used to determine the statistical significance between DMSO (control) and juglone treatment, nested one way-ANOVA was used to determine the statistical significance between wild-type and mutant strains. $P$ values are indicated with asterisks *$p < 0.05$, **$p < 0.01$, ***$p < 0.001$, or number symbols #$p < 0.05$, ##$p < 0.01$, ###$p < 0.001$. Error bars in the figures indicate the standard error of the mean (±SEM). Bar graph and Box-and-whisker plots were generated on GraphPad Prism 8. Sample sizes for coelomocyte assays were 20–30 animals per condition, for toxicity assays were 100–200 animals, for mito-HyPer and GCaMP imaging were 20–30.

**Reporting summary.** Further information on research design is available in the Nature Research Reporting Summary linked to this article.

## Data availability
All data supporting the findings of this study are available within the paper and its supplementary files. Source data are provided with this paper.

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

## Acknowledgements

*C. elegans* strains used in this work were provided by the Caenorhabditis Genetics Centre (Univ. of Minnesota), which is funded by the NIH National Center for Research Resources (NCRR). Thanks to Trisha Staab for the initial observation that *egl-3* mutants are hypersensitive to juglone toxicity, to the Lillian Schoofs lab for testing the interaction between FLP-1 and NPR-4, to Karen Chang for providing the HyPer construct, and to Eric Jorgensen for providing the UNC-116::TOM-7 constructs. Thanks to members of the Sieburth lab for discussions and critical reading of the manuscript. This work was supported by the Southern California Environmental Health Sciences Center, grant #P30ES007048 and National Institutes of Health (NIH), grant #NS099414 to D.S.

## Author contributions

Q.J. designed and preformed experiments, analyzed data and wrote the manuscript. D.S. designed experiments and edited the manuscript.

## Competing interests

The authors declare no competing interests.

## Additional information

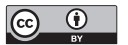

