## [Peer Review File · Nature Communications]

REVIEWER COMMENTS

Reviewer #1 (Remarks to the Author):

Using the *C. elegans* AIY neurons as a model system, Jia and Sieburth find that mitochondria derived H₂O₂ can serve as a signaling molecule in response to environmental oxidative stress to elicit neuropeptide FLP-1 secretion, which further activates the transcription factor SKN-1/Nrf2 in distal tissues and protects animals from ROS-mediated toxicity. The secretion of FLP-1 is dependent on cysteine sulfenylation of the calcium independent PKC family member PKC-1. Their work also suggests that bacterial-derived signals from diet could lead to changes in mtH₂O₂ production, which further regulates neuropeptide secretion.

This is an exceptionally thorough and interesting manuscript. The experiments were well designed, the data are clearly presented, and the manuscript is well written. Their work is of broad interest in the field, and opens up many intriguing questions. I recommend publication essentially as is. I have only minor suggestions to improve this manuscript.

1) The data in this paper show that juglone treatment significantly increases mitochondrial calcium levels, and that mitochondria calcium influx is required for mtH₂O₂ production. Is the idea that the mitochondria calcium influx the most upstream event known so far? If so, how is calcium influx regulated to respond to oxidative stress? Or is it possible that juglone treatment directly influences the activity of SOD-2, and that calcium influx facilitates the production of mtH₂O₂? Some discussion of the potential sequence of the events in terms of mitochondria sensing oxidative stress would be helpful. Given the extensive mechanistic detail already contained in the manuscript, I feel that this discussion can be speculative and does not need experimental support.

2) It is interesting that unlike *flp-1* single mutants, *flp-1; flp-18* double displayed normal sensitivity to Juglone (Figure S4H). Is there a potential explanation for this?

3) The result of *ric-7*+H₂O₂ treatment in Figure 6C is not cited in the main text. I think this is an important result, indicating that although other aspects of mitochondria such as electron transport chain function and calcium influx are essential for FLP-1 secretion (as in Figure S2B & Figure 5), adding back H₂O₂ in the absence of axonal mitochondria is sufficient to elicit FLP-1 secretion, which further supports the idea that mtH₂O₂ in close proximity to DCV release sites is essential for FLP-1 release.

4) In Figure 4B, juglone treatments in *pkc-1* and *unc-31* mutants seem to result in a low ratiometric measurement for H₂O₂ in the cytoplasm surrounding mitochondria (black areas surrounding mitochondria). Does this suggest local depletion of H₂O₂ in the neighborhood of mitochondria in those mutants? Is this readout due to leak of mtHyper into the cytoplasm?

5) Page 38 Line 1271-1275 says "For mito-Hyper fluorescence quantification, Z stacks were obtained using CFP420, and GFP516 filter sets sequentially." It looks like the excitation wavelength is cited for CFP and the emission for GFP. Please clarify.

6) In Figure 8F, the interesting result that the *flp-1* mutant fed with HT115 is more sensitive to juglone treatment compared to *flp-1* fed with OP50 is not discussed. Can you speculate why this is the case?

Grammar errors:

1) Line 750 – 755: "Interestingly, the function we found for *pdrx-3* in inhibiting H₂O₂ signaling in contrasts with that of the cytoplasmic peroxiredoxin, *prdx-2*, which promotes H₂O₂ signaling by functioning as a redox-active "relay" protein for H₂O₂ in sensory neurons during sensory transduction

(BHATLA AND HORVITZ 2015; LI et al. 2016), revealing distinct mechanisms by which cytoplasmic vs. mitochondrial peroxiredoxins impact H₂O₂ signaling in *C. elegans*.”

Remove 'in' in the phrase "in contrasts"

2) Line 781 – 783: "Therefore, we propose that H₂O₂-induced C524 sulfenylation promotes membrane recruitment of PKC-1 to DCV release sites, were it facilitates calcium-dependent FLP-1 release"

"were" should be "where"

Reviewer #2 (Remarks to the Author):

Review of Jia and Sieburth, Nature comms

This MS describes a role for neuronal mitochondria in regulation of release of neuropeptides in the *C. elegans* nervous system. The authors investigated the basis for juglone-hypersensitivity of certain neuropeptide processing mutants and trace this to release of FLP-1 neuropeptides from a pair of interneurons. The authors use a variety of lines of evidence to show that mitochondrial H₂O₂ likely triggers local neuropeptide release, which then acts distally on intestinal epithelial cells to regulate expression of anti-oxidant pathways.

Overall the work is extremely thorough and shows extensive evidence for the proposed pathway, which may be normally triggered by different bacterial diets or environmental toxins. The manuscript is well written although quite dense reading given the large number of experiments. The only major concern is the reliance on some reporter gene based assays whose physiological relevance may not be completely established. For example much use is made of quantitating FLP-1-Venus accumulation in coelomocytes but it is not clear what different levels of accumulation are measuring. The baseline for this assay also seems quite variable. The discussion could be improved by consideration of the limitations of such assays as proxies for neuropeptide secretion. There are some other minor technical issues which could be addressed by improved documentation.

Minor points

Which AIY-FLP-1-Venus transgenes are being used in which experiments? Multiple transgenes are listed, but not cross referenced to the data. A table showing which transgene is used in which experiment would be helpful.

Images generally lack scale bars and the ROIs used in quantitation are not clear—an asterisk is used to indicate the general area but not the ROI itself.

A raw data file should be required for the quantitative fluorescence measurements, which are notoriously sensitive to animal growth and imaging conditions. In fact a raw data file would be helpful for all the experimental results displayed.

Please clarify what kind of blue light source was used.

Statistics throughout use a t-test, which assumes data are normally distributed and does not correct for multiple comparisons. The authors should justify their use of statistical tests.

Reviewer #3 (Remarks to the Author):

“Mitochondrial hydrogen peroxide positively regulates neuropeptide secretion during diet-induced activation of the oxidative stress response” by Qi Jia and Derek Sieburth.

In this manuscript, authors showed how environmental stress, particularly ROS, is sensed in the nervous system to induce ROS resistance. To investigate the tolerance to ROS, they focused on juglone resistance. Although juglone has multiple effects on cells and living organisms including antioxidant properties (Ahmed and Suzuki, 2019), in *C. elegans* it is considered to cause toxicity by generating oxidative stress. Gene hunting and rescue experiments revealed that the neuropeptide FLP-1 released from AIY and its receptor NPR-4 in the intestine have roles in juglone resistance. The authors further elucidated that hydrogen peroxide, produced from ROS in AIY, stimulates the release of FLP-1. Interestingly, it was also shown that the release of FLP-1 is regulated by the (possibly direct) modification of a cysteine residue of PKC-1 in response to increased hydrogen peroxide. In addition, authors showed that FLP-1 signaling varied not only in response to the xenobiotic agent juglone, but also to the growth environment of the nematode, i.e., different *E. coli* strains.

It should be praised that authors performed such a comprehensive work. The results are very interesting and most of the authors' claims are strongly supported by the experimental data. However, some problems remain and some additional experiments and discussions are needed to complete the work.

Major Points.

1. The biological significance of the signaling pathways identified in this study is summarized in Fig. 8G and authors propose that AIY is the biological sensor of oxidative stress conditions, such as those caused by bacterial food. This interpretation is understandable if we are talking about systemic stress, but in the case of food, it is puzzling because food is consumed by the intestine, but in Fig. 8G the neuron jumps in to sense the bacteria and signal back to the intestine. Then the question that many of the readers may ask arises. What is AIY sensing? It could be one of the following (as discussed in Discussion): i) oxidants included in bacteria, ii) other metabolite of the bacteria that leads to generation of oxidative stress in AIY, iii) sensation of bacteria by the sensory neurons. Authors need to add some more information to support the model. i) can be tested by monitoring the H₂O₂ level by HyPer in other cells that neighbor AIY. iii) can be tested by monitoring AIY FLP-1 secretion in *osm-6* mutants or *unc-13* mutants in response to OP50/HT115.

2. The finding of the role of *pkc-1* is one of the major achievement in this study. As discussed in Discussion, the observation that only FLP-1 but not FLP-18 is regulated by oxidative stress may be because *pkc-1* regulates only FLP-1-including DCV. This was suggested by not tested. A simple test needs to be included within the context of this manuscript, namely comparing the AIY FLP-18::Venus secretion in *pkc-1* mutants with that in the wild type.

3. In figure S1D, *flp-1* mutants are not sensitive to paraquat which generates ROS. Why is that? Authors need to comment on this in the main text (currently paraquat is not mentioned at all). It is understood that FLP-1 signal carries only a part of the resistance response, because *flp-1* mutants have only a partial defect in the resistance to juglone. Therefore, it is conceivable that the contribution of FLP-1 is even smaller in the case of paraquat, because, for example, its direct effect on the body is so strong. In that scenario, authors may need to look at AIY FLP-1 secretion in response to paraquat to show that FLP-1 does respond to paraquat because it generates oxidative stress.

Minor points

1. Related to major point 1, in Fig. 8F, y axis label says “normalized”. What does this mean? Were both OP50 and HT115 normalized independent of each other, or all the bars were normalized using a common scaling factor? Authors need to show raw values. It is important for the consideration on the role of stress response in natural environments. HT115 induces a higher basal level of FLP-1 secretion and probably activates intestinal SKN-1, which is expected to cause higher resistance in HT115-raised

worms to the insult by juglone. Is this the case or not? In any case, this piece of information needs to be disclosed.

2. In Fig. 4E, the "sod-2 +H₂O₂" bar shows n.s., and "unc-31+H₂O₂" bar shows ***. Please state in the figure legend what are compared in these statistical tests.

3. In all figures, it is generally recommended to use dot representation showing each measurement, rather than only mean and s.e.m. It would be a more honest representation of the variation in the data.

4. page 3, line 64: "such" probably needs to read "such as".

5. page 11, line 310: "into to" must be a typo of "into".

6. page 18, line 527: "sulfenylation" -> "sulfenylation"

7. page 18, line 534: "relies of" -> "relies on"

8. page 21, line 620: "Figure 8B and B" probably needs to read "Figure 8B and C"

9. page 22, line 659: The subject suddenly changes from C. elegans to mammalian brain. It would be better to add some words to indicate the change for the sake of readability.

10. page 23, line 694: "by" before "glucose" is not necessary.

11. page 24, line 706: "by"s before "phosphorylation" and "ROS" are not necessary.

REVIEWER COMMENTS

Reviewer #1 (Remarks to the Author):

Using the *C. elegans* AIY neurons as a model system, Jia and Sieburth find that mitochondria derived H₂O₂ can serve as a signaling molecule in response to environmental oxidative stress to elicit neuropeptide FLP-1 secretion, which further activates the transcription factor SKN-1/Nrf2 in distal tissues and protects animals from ROS-mediated toxicity. The secretion of FLP-1 is dependent on cysteine sulfenylation of the calcium independent PKC family member PKC-1. Their work also suggests that bacterial-derived signals from diet could lead to changes in mtH₂O₂ production, which further regulates neuropeptide secretion.

This is an exceptionally thorough and interesting manuscript. The experiments were well designed, the data are clearly presented, and the manuscript is well written. Their work is of broad interest in the field, and opens up many intriguing questions. I recommend publication essentially as is. I have only minor suggestions to improve this manuscript.

1) The data in this paper show that juglone treatment significantly increases mitochondrial calcium levels, and that mitochondria calcium influx is required for mtH₂O₂ production. Is the idea that the mitochondria calcium influx the most upstream event known so far? If so, how is calcium influx regulated to respond to oxidative stress? Or is it possible that juglone treatment directly influences the activity of SOD-2, and that calcium influx facilitates the production of mtH₂O₂? Some discussion of the potential sequence of the events in terms of mitochondria sensing oxidative stress would be helpful. Given the extensive mechanistic detail already contained in the manuscript, I feel that this discussion can be speculative and does not need experimental support.

**This is an interesting question. We have added to the Discussion the following speculation:
Page 24:**

Third, H₂O₂ levels could be regulated by controlling calcium influx into mitochondria. Mitochondrial calcium is an activator the ETC, and sustained ETC activation can lead to mtROS production (WILLIAMS et al. 2015). In this scenario, ROS generated by oxidants would increase calcium levels in AIY mitochondria, possibly by regulating calcium influx through MCU1. Consistent with this idea, we found that calcium entry through MCU-1 was necessary for juglone-induced FLP-1 secretion. The activity of the MCU channel is proposed to be modulated by a number of post-transcriptional modifications, including by phosphorylation, and by ROS mediated S-glutathionylation (MAMMUCARI et al. 2017). We also found that mitochondria calcium influx is required for juglone-induced mtH₂O₂ production and for mitochondrial calcium increases, suggesting that mitochondrial calcium may also facilitate the generation of H₂O₂ in response to oxidants, which might in turn lead to increased calcium levels in mitochondria. Determining the sequence of events relating to mtH₂O₂ production and calcium influx following oxidant exposure will help to clarify the mechanism by which mitochondria sense oxidative stress.

2) It is interesting that unlike flp-1 single mutants, flp-1; flp-18 double displayed normal sensitivity to Juglone (Figure S4H). Is there a potential explanation for this?

**We have added a sentence to the Discussion to address this:
Page 23:**

Interestingly, we found that *flp-1* mutants were sensitive to juglone-induced toxicity, whereas *flp-18* mutants were resistant to juglone-induced toxicity, and *flp-1; flp-18* double mutants exhibited an intermediate juglone response (Figure S4I), suggesting a role for *flp-18* in inhibiting the oxidative stress response by antagonizing the protective effects of *flp-1* signaling.

3) The result of *ric-7*+H₂O₂ treatment in Figure 6C is not cited in the main text. I think this is an important result, indicating that although other aspects of mitochondria such as electron transport chain function and calcium influx are essential for FLP-1 secretion (as in Figure S2B & Figure 5), adding back H₂O₂ in the absence of axonal mitochondria is sufficient to elicit FLP-1 secretion, which further supports the idea that mtH₂O₂ in close proximity to DCV release sites is essential for FLP-1 release.

We have added a discussion of this experiment in the Results:

Page 17:

Importantly, we found that the increase in FLP-1::Venus secretion caused by mito-miniSOG activation was blocked by *ric-7* mutations (Figure 4D). However, *ric-7* mutations failed to block H₂O₂-induced FLP-1::Venus secretion (Figure 6C), indicating that H₂O₂ can bypass the need for axonal mitochondria. These results further support the idea that mtH₂O₂ generated locally at DCV release sites in axons is essential for inducing FLP-1 release.

4) In Figure 4B, juglone treatments in *pkc-1* and *unc-31* mutants seem to result in a low ratiometric measurement for H₂O₂ in the cytoplasm surrounding mitochondria (black areas surrounding mitochondria). Does this suggest local depletion of H₂O₂ in the neighborhood of mitochondria in those mutants? Is this readout due to leak of mtHyper into the cytoplasm?

We thank the reviewer for noticing this error. The “halos” around the mitochondria seen in those panels were not representative and seemed to arise from false coloring when the figure panels were prepared. We have replaced the *pkc-1* and *unc-31* images in Fig. 4B with new representative images.

We have also added to Fig 4B mito hyper data for the *prdx-3* mutant, which confirms that *prdx-3* negatively regulates peroxide levels in AIY axonal mitochondria.

Page 15:

trx-2 or *prdx-3* mutants also exhibited increased mito-HyPer punctal fluorescence intensity in AIY compared to wild type controls (Figure 4B).

B

5) Page 38 Line 1271-1275 says “For mito-HyPer fluorescence quantification, Z stacks were obtained using CFP420, and GFP516 filter sets sequentially.” It looks like the excitation wavelength is cited for CFP and the emission for GFP. Please clarify.

We have corrected this in the Methods.

Page 41:

Z stacks were obtained using GFP (excitation/emission: 450nm/520nm) and CFP (excitation/emission: 420/480nm) filter sets sequentially.

6) In Figure 8F, the interesting result that the flp-1 mutant fed with HT115 is more sensitive to juglone treatment compared to flp-1 fed with OP50 is not discussed. Can you speculate why this is the case?

We agree that there is a modest difference at the concentration of juglone used (120µM)> However, a more comprehensive dose-response analysis would be needed to address whether HT115 indeed makes flp-1 mutants more sensitive to juglone toxicity. Since the figure is meant to show that flp-1 mutants are more sensitive to toxicity compared to wild type controls regardless of which food source is used, we feel that commenting on differences between the food sources may be premature until a more complete analysis is done.

Grammar errors:

1) Line 750 – 755: “Interestingly, the function we found for prdx-3 in inhibiting H2O2 signaling in contrasts with that of the cytoplasmic peroxiredoxin, prdx-2, which promotes H2O2 signaling by functioning as a redox-active “relay” protein for H2O2 in sensory neurons during sensory transduction (BHATLA AND HORVITZ 2015; LI et al. 2016), revealing distinct mechanisms by which cytoplasmic vs. mitochondrial peroxiredoxins impact H2O2 signaling in C. elegans.”

Remove ‘in’ in the phrase “in contrasts”

Corrected.

2) Line 781 – 783: “Therefore, we propose that H2O2-induced C524 sulfenylation promotes membrane recruitment of PKC-1 to DCV release sites, were it facilitates calcium-dependent FLP-1 release”
“were” should be “where”

Corrected.

Reviewer #2 (Remarks to the Author):

Review of Jia and Sieburth, Nature comms

This MS describes a role for neuronal mitochondria in regulation of release of neuropeptides in the C. elegans nervous system. The authors investigated the basis for juglone-hypersensitivity of certain neuropeptide processing mutants and trace this to release of FLP-1 neuropeptides from a pair of interneurons. The authors use a variety of lines of evidence to show that mitochondrial H2O2 likely triggers local neuropeptide release, which then acts distally on intestinal epithelial cells to regulate expression of anti-oxidant pathways.

Overall the work is extremely thorough and shows extensive evidence for the proposed pathway, which may be normally triggered by different bacterial diets or environmental toxins. The manuscript is well written although quite dense reading given the large number of experiments. The only major concern is

the reliance on some reporter gene based assays whose physiological relevance may not be completely established. For example much use is made of quantitating FLP-1-Venus accumulation in coelomocytes but it is not clear what different levels of accumulation are measuring. The baseline for this assay also seems quite variable. The discussion could be improved by consideration of the limitations of such assays as proxies for neuropeptide secretion. There are some other minor technical issues which could be addressed by improved documentation.

We agree that the coelomocyte assay is not a direct measurement of neuropeptide release, but we believe that it is a good proxy for quantifying FLP-1 release since we have shown that ROS do not alter constitutive secretion (Fig. S4G), and manipulations that are known to reduce or increase neuropeptide secretion cause corresponding reductions or increases in coelomocyte fluorescence of FLP-1::Venus (Fig 2F and 2G), respectively. The coelomocyte uptake assay has been used in multiple studies by several laboratories to analyze neuropeptide secretion, and it is generally recognized as an appropriate readout for neuropeptide release that corresponds well with electron microscopy studies of DCV abundance at release sites. Moreover, changes in FLP-1 secretion we report here strongly correlate with changes in *flp-1*-regulated responses such as toxicity and SKN-1 activation. Accordingly, we consider these responses to be independent measures for endogenous FLP-1 secretion.

Minor points

Which AIY-FLP-1-Venus transgenes are being used in which experiments? Multiple transgenes are listed, but not cross referenced to the data. A table showing which transgene is used in which experiment would be helpful.

We have amended the supplemental table and we now indicate which transgenes were used for which experiment in Methods.

Page 41:

All FLP-1::Venus secretion assays from AIY neurons were performed using the *vjls150* transgene, which is integrated on LG III, except for *trxr-2* mutants, which were analyzed using *vjls152*, which is integrated on LG I.

Images generally lack scale bars and the ROIs used in quantitation are not clear—an asterisk is used to indicate the general area but not the ROI itself.

We have added scale bars to all panels. We have added to Methods a description of how *Pgst-4::GFP* fluorescence intensity was calculated.

Page 41:

For quantification of *Pgst-4::gfp* expression, a 16-pixel diameter circle was drawn in the posterior intestine (ROI) and the average fluorescence intensity within the area of the circle was calculated using MetaMorph. Background fluorescence was measured as the average intensity within a same-sized circle positioned next to the animal (coverslip fluorescence) and this value was subtracted from the ROI fluorescence to generate the fluorescence intensity value.

A raw data file should be required for the quantitative fluorescence measurements, which are notoriously sensitive to animal growth and imaging conditions. In fact a raw data file would be helpful for all the experimental results displayed.

We have included all the raw data for each figure as an excel file.

Please clarify what kind of blue light source was used.

We have modified the Methods to clarify this.

Page 37:

For miniSOG-induced cell ablation, an EXFO mercury arc lamp was used as the blue light source. 30-40 L4 animals were transferred onto fresh NGM plates with OP50 and animals were exposed to continuous 50mW/cm² blue light for 30min and recovered at 20°C in dark for 16 hours before toxicity assays. For miniSOG-induced ROS generation, LED light with pre-built MSOG0001 filter module (TriTech Research) was used as the blue light source. L4 stage worms were transferred onto fresh NGM plates with OP50 and animals were exposed to continuous 100mW/cm² blue light for 1min and recovered at 20°C in dark for 10min before taking images.

Statistics throughout use a t-test, which assumes data are normally distributed and does not correct for multiple comparisons. The authors should justify their use of statistical tests.

We have added to Methods the following description of the t-tests used:

Page 41:

Statistical analysis, Bar graphs, and Box-and-whisker plots were generated using GraphPad Prism 9. Unpaired t-test (two tails) was used to determine the statistical significance between DMSO (control) and juglone treatment, nested one way-ANOVA was used to determine the statistical significance between wild type and mutant strains. P values less than 0.05, 0.01 or 0.001 are indicated with asterisks * (p < 0.05), ** (p < 0.01), *** (p < 0.001), or number symbols # (p < 0.05), ## (p < 0.01), ### (p < 0.001). Error bars in the figures indicate the standard error of the mean (\pm SEM). The exact sample sizes (n) and \pm SEM values corresponding to all Figures are listed in each Figure legend.

Reviewer #3 (Remarks to the Author):

“Mitochondrial hydrogen peroxide positively regulates neuropeptide secretion during diet-induced activation of the oxidative stress response” by Qi Jia and Derek Sieburth.

In this manuscript, authors showed how environmental stress, particularly ROS, is sensed in the nervous system to induce ROS resistance. To investigate the tolerance to ROS, they focused on juglone resistance. Although juglone has multiple effects on cells and living organisms including antioxidant properties (Ahmed and Suzuki, 2019), in *C. elegans* it is considered to cause toxicity by generating oxidative stress. Gene hunting and rescue experiments revealed that the neuropeptide FLP-1 released from AIY and its receptor NPR-4 in the intestine have roles in juglone resistance. The authors further elucidated that hydrogen peroxide, produced from ROS in AIY, stimulates the release of FLP-1. Interestingly, it was also shown that the release of FLP-1 is regulated by the (possibly direct) modification of a cysteine residue of PKC-1 in response to increased hydrogen peroxide. In addition, authors showed that FLP-1 signaling varied not only in response to the xenobiotic agent juglone, but also to the growth environment of the nematode, i.e., different *E. coli* strains.

It should be praised that authors performed such a comprehensive work. The results are very interesting and most of the authors' claims are strongly supported by the experimental data. However, some problems remain and some additional experiments and discussions are needed to complete the work.

Major Points.

1. The biological significance of the signaling pathways identified in this study is summarized in Fig. 8G and authors propose that AIY is the biological sensor of oxidative stress conditions, such as those caused

by bacterial food. This interpretation is understandable if we are talking about systemic stress, but in the case of food, it is puzzling because food is consumed by the intestine, but in Fig. 8G the neuron jumps in to sense the bacteria and signal back to the intestine. Then the question that many of the readers may ask arises. What is AIY sensing? It could be one of the following (as discussed in Discussion): i) oxidants included in bacteria, ii) other metabolite of the bacteria that leads to generation of oxidative stress in AIY, iii) sensation of bacteria by the sensory neurons. Authors need to add some more information to support the model. i) can be tested by monitoring the H₂O₂ level by HyPer in other cells that neighbor AIY. iii) can be tested by monitoring AIY

FLP-1 secretion in *osm-6* mutants or *unc-13* mutants in response to OP50/HT115.

We agree with the reviewer that there are a number of possible sources for the signal(s) that AIY senses that regulate FLP-1 secretion. We have modified Figure 8G to include the possible inputs into AIY that the reviewer proposes here and that we discussed in the Discussion. For point i), we have not measured peroxide levels in neighboring cells, but we have shown that ROS-induced neuropeptide secretion does not occur in the nearby AVK neuron (Fig. S4M) or the cholinergic motor neurons in the ventral nerve cord (Fig. S4O), suggesting that the effects of ROS on neuropeptide secretion are not global and are likely to be restricted to AIY and possibly other neurons/tissues that have not been identified. For point iii) we have shown that *osm-6* mutants respond to juglone treatment, indicating that sensory transduction is not likely to mediate the effects of juglone, at least for animals grown on OP50. In addition, FLP-1 is secreted in animals grown of either OP50 or HT115, making it likely that FLP-1 will still be secreted in *osm-6* mutants following HT115 feeding. Determining whether the effects of HT115 are mediated by sensory transduction or not is an interesting question, but we feel that to address it here would be beyond the scope of this study.

2. The finding of the role of *pkc-1* is one of the major achievement in this study. As discussed in Discussion, the observation that only FLP-1 but not FLP-18 is regulated by oxidative stress may be because *pkc-1* regulates only FLP-1-including DCV. This was suggested by not tested. A simple test needs to be included within the context of this manuscript, namely comparing the AIY FLP-18::Venus secretion in *pkc-1* mutants with that in the wild type.

We have added Fig. S4K showing that *pkc-1* mutations do not alter FLP-18::Venus secretion from AIY.

Page 17, Results:

We found that *pkc-1* null mutations significantly reduced FLP-1::Venus secretion from AIY (Figure 7B) without altering FLP-18::Venus secretion (Figure S4K).

Page 23, Discussion:

Our observation that FLP-1 but not FLP-18 secretion from AIY is increased by juglone and by *pkc-1*, suggests that FLP-1 and FLP-18 secretion may be differentially regulated, raising the possibility that these peptides are packaged into distinct DCV pools that may differ in their proximity to mitochondria, or their ability to be regulated by PKC-1 signaling.

J

3. In figure S1D, flp-1 mutants are not sensitive to paraquat which generates ROS. Why is that? Authors need to comment on this in the main text (currently paraquat is not mentioned at all). It is understood that FLP-1 signal carries only a part of the resistance response, because flp-1 mutants have only a partial defect in the resistance to juglone. Therefore, it is conceivable that the contribution of FLP-1 is even smaller in the case of paraquat, because, for example, its direct effect on the body is so strong. In that scenario, authors may need to look at AIY FLP-1 secretion in response to paraquat to show that FLP-1 does respond to paraquat because it generates oxidative stress.

As requested by the reviewer, we examined the effect of paraquat on FLP-1 secretion. We found that 10 min paraquat treatment did not result in an increase in FLP-1:Venus coelomocyte fluorescence (see data below). Since we don't know whether paraquat was able to enter the animals in 10 minutes, we cannot make a conclusion about how paraquat affects FLP-1 secretion from this negative result. For this reason, we prefer not to include the paraquat data or to comment on the effects of paraquat on FLP-1 secretion in this study.

B

We added Fig S4E showing that 10 min sodium arsenite treatment increase FLP-1::Venus coelomocyte fluorescence, which independently confirms that acute oxidant exposure increases FLP-1 secretion.

Page 11, Results:

Similarly, a 10 minute treatment with the oxidant sodium arsenite significantly increased FLP-1::Venus secretion compared to untreated controls (Figure S4E).

C

Minor points

1. Related to major point 1, in Fig. 8F, y axis label says “normalized”. What does this mean? Were both OP50 and HT115 normalized independent of each other, or all the bars were normalized using a common scaling factor? Authors need to show raw values. It is important for the consideration on the role of stress response in natural environments. HT115 induces a higher basal level of FLP-1 secretion and probably activates intestinal SKN-1, which is expected to cause higher resistance in HT115-raised worms to the insult by juglone. Is this the case or not? In any case, this piece of information needs to be disclosed.

We did not directly compare OP50 and HT115 on survival since we refer to the published data showing greater survival in response to oxidants including juglone when animals are cultured on HT115 compared to OP50. Fig 8F shows that *flp-1* signaling is important for survival on either food source. We have clarified how normalization was done in the Figure legend.

Page 35:

Survival of *flp-1* mutants was normalized to wild type animals raised on the same bacterial strain.

2. In Fig. 4E, the “*sod-2* +H₂O₂” bar shows n.s., and “*unc-31*+H₂O₂” bar shows ***. Please state in the figure legend what are compared in these statistical tests.

Replaced with updated Fig 4E containing symbols that distinguish the different comparisons:

E

3. In all figures, it is generally recommended to use dot representation showing each measurement, rather than only mean and s.e.m. It would be a more honest representation of the variation in the data.

All data has been replaced with box-and-whisker plots. An example is shown below.

F

4. page 3, line 64: “such” probably needs to read “such as”.

5. page 11, line 310: “into to” must be a typo of “into”.

6. page 18, line 527: “sylfenylation” -> “sulfenylation”

7. page 18, line 534: “relies of” -> “relies on”

8. page 21, line 620: “Figure 8B and B” probably needs to read “Figure 8B and C”

9. page 22, line 659: The subject suddenly changes from *C. elegans* to mammalian brain. It would be better to add some words to indicate the change for the sake of readability.

Corrected to:

Neurons are among the most metabolically active cells, yet in mammals, mature neurons possess a limited capacity to neutralize ROS since Nrf2 activity is weak.

10. page 23, line 694: “by” before “glucose” is not necessary.

11. page 24, line 706: “by”s before “phosphorylation” and “ROS” are not necessary.

All corrected.

REVIEWERS' COMMENTS

Reviewer #1 (Remarks to the Author):

The authors have addressed all my comments and those of the other reviewers. The revised manuscript is extremely thorough and is ready for publication. This is an excellent and very interesting study.

Reviewer #2 (Remarks to the Author):

The revised MS has addressed my concerns and I support publication.

Reviewer #3 (Remarks to the Author):

This is a very interesting paper and upon revision the concerns have been mostly addressed. The question of how food and stress signals are conveyed to AIY is still not fully answered, but because the major conclusion of the study is that AIY sends stress response signal via FLP-1, addressing the question is probably beyond the scope of the study.

Only minor corrections are requested at this stage.

1. In page 23 line 689, "regulated" and "raising" are fused.
2. Figure 8F legend says n=5080 animals in triplicate. This is a quite ambiguous description. The figure indicates three dots for each of four conditions. Is 5080 the total number of animals counted for the whole experiment? Authors need to indicate numbers of animals for each condition (for example n=xxxx for flp-1 fed HT115). Because survival rate is lower for OP50, authors may have counted more animals for OP50. It is important for judgement of the reliability of the data, especially because authors decided not to show the raw survival rate.

"Mitochondrial hydrogen peroxide positively regulates neuropeptide secretion during diet-induced activation of the oxidative stress response" (NCOMMS-20-35629).

Below, please find our responses to Reviewers' corrections **in red**:

Reviewer #3 (Remarks to the Author):

This is a very interesting paper and upon revision the concerns have been mostly addressed. The question of how food and stress signals are conveyed to AIY is still not fully answered, but because the major conclusion of the study is that AIY sends stress response signal via FLP-1, addressing the question is probably beyond the scope of the study.

Only minor corrections are requested at this stage.

1. In page 23 line 689, "regulated" and "raising" are fused. **Corrected in manuscript**
2. Figure 8F legend says n=5080 animals in triplicate. This is a quite ambiguous description. The figure indicates three dots for each of four conditions. Is 5080 the total number of animals counted for the whole experiment? Authors need to indicate numbers of animals for each condition (for example n=xxxx for flp-1 fed HT115). Because survival rate is lower for OP50, authors may have counted more animals for OP50. It is important for judgement of the reliability of the data, especially because authors decided not to show the raw survival rate. **Corrected in manuscript.**